# Feature-specific neural reactivation during episodic memory

Michael B. Bone [1,2✉], Fahad Ahmad[1] & Bradley R. Buchsbaum [1,2]

We present a multi-voxel analytical approach, feature-specific informational connectivity (FSIC), that leverages hierarchical representations from a neural network to decode neural reactivation in *f*MRI data collected while participants performed an episodic visual recall task. We show that neural reactivation associated with low-level (e.g. edges), high-level (e.g. facial features), and semantic (e.g. "terrier") features occur throughout the dorsal and ventral visual streams and extend into the frontal cortex. Moreover, we show that reactivation of both low- and high-level features correlate with the vividness of the memory, whereas only reactivation of low-level features correlates with recognition accuracy when the lure and target images are semantically similar. In addition to demonstrating the utility of FSIC for mapping feature-specific reactivation, these findings resolve the contributions of low- and high-level features to the vividness of visual memories and challenge a strict interpretation the posterior-to-anterior visual hierarchy.

[1] Rotman Research Institute at Baycrest, Toronto, ON M6A 2E1, Canada. [2] Department of Psychology, University of Toronto, Toronto, ON M5S 1A1, Canada. ✉email: michael.bone@mail.utoronto.ca

Not all of our conscious memories for past events have the same quality of experience: some are vague and fuzzy, while others are sharp and detailed—sometimes nearly on par with the "fidelity" of direct perceptual experience. What accounts for this variability in the sharpness and "resolution" of memories? Researchers studying mental imagery, episodic memory, and working memory have converged on the idea that memories are constructed from the same neural representations that underlie direct perception[1–7], a process known as neural reactivation[8,9]. Researchers have found that measures of neural reactivation throughout the dorsal and ventral visual streams reflect the content of episodic memory[4,5,7,10,11], including low-level image properties such as edge orientation and luminosity[6,12,13], as well as high-level semantic properties[14,15]. Moreover, the degree of neural reactivation is correlated with memory vividness[16–20].

The parallels between perception and memory extend beyond the representational overlap within posterior visual regions. As with perception, visual memory is subject to capacity constraints[21], and depends on similar executive processes, such as selective attention[20,22–25] and working memory[26–28]. These executive processes serve to enhance and maintain neural reactivation of task-relevant image features within posterior visual regions via top-down projections from the frontal cortex[29–34].

Although there is strong evidence that a network of frontal cortical areas contributes to visual memory, there is a debate over the nature of the representations within these regions. By one account, frontoparietal regions encode abstract task-level representations such as category membership[35–38], rules, and stimulus-response mappings[39]. However, stimulus-specific responses have also been discovered within prefrontal regions[18,40–43], with some areas of the frontal cortex supporting both task-general and stimulus-specific representations in a high-dimensional state space[44–46]. Whereas evidence for stimulus-specific representations within the frontal cortex has been growing over the last decade, there is still little information about the granularity of sensory features represented in the frontal cortex, as the tools for detecting such representations are just beginning to emerge.

The detection of feature-specific neural representations has advanced significantly over the past few years with the advent of brain-inspired deep convolutional neural networks[47] (CNN). Early attempts at identifying and localizing neural activity associated with specific visual features focused on either high-level sematic/categorical features[14,48–52], low-level features such as edges[6,53] or both[54,55]—limiting findings to a small portion of the cortical visual hierarchy. In contrast, features extracted from the layers of a deep CNN have been linked to activity over nearly the entire visual cortex during perception, with a correspondence between the hierarchical structures of the CNN and cortex[56–60].

The architecture of feed-forward CNNs is such that features from higher layers of the network are composed of features from lower layers, resulting in strong inter-layer correlations. Thus, any method that does not control for these inter-layer correlations will be prone to falsely detect reactivation of features from (nearly) all levels of the visual hierarchy when only a small subset of the feature-levels are present within a given brain region. Güçlü and van Gerven[57] and Seeliger et al.[60] developed a method to address this issue that first assigns the layer that best predicts a given voxel/source's activity to that voxel/source, and then uses the proportion of voxel/sources assigned to each layer within a region of interest (ROI) to infer the feature-levels represented within that cortical region. This approach, however, may overlook feature-levels that are weakly represented within a given region, due to the simplifying assumption that only one feature level is represented per voxel/source.

To overcome some of these previous limitations in identifying feature-specific reactivation during memory recall, we introduce feature-specific informational connectivity (FSIC), a measure that incorporates a voxel-wise modeling and decoding approach[6], coupled with a variant of informational connectivity[61,62]. Our method exploits trial-by-trial variability in the retrieval of episodic memories by measuring the synchronized shifts in reactivation across cortical regions. We demonstrate that this approach identifies feature-specific reactivation while accounting for inter-layer correlations and retaining sensitivity to more weakly represented features.

We use FSIC to examine feature-specific reactivation across the neocortex during a task where subjects recall and visualize naturalistic images. The experiment has two video viewing runs, used to train the encoding models, and three sets of alternating encoding and retrieval runs (Fig. 1a). During encoding runs participants memorize a sequence of color images while performing a 1-back task. In the following retrieval runs, participants' recall and recognition memory of the images are assessed. Feature-specific neural reactivation is measured while participants visualize a cued image within a light-gray rectangle, followed by a memory vividness rating. An image is then presented that is either identical to the cued image or a similar lure, and the participants judge whether they had seen the image during encoding, followed by a rating of their confidence in this response.

Given the purported role of the frontal cortex in coordinating visual representations within posterior sensory regions[29–34], we hypothesize that neural reactivation for all visual feature-levels should occur within—and be synchronized between—these cortical regions. Beyond establishing the cortical distribution of feature-specific visual representations, we are also interested in their connection to memory performance. To this end, we investigate the relationship between feature-specific reactivation during recall and both subjective (vividness ratings) and objective (recognition accuracy) behavioral memory measures. We hypothesize that reactivation of all feature levels will correlate with the vividness of the recalled image, and that lower level representations will have the strongest correlation because these features are most clearly associated with the phenomenology of vivid memories[22,63]. We also hypothesize that during recognition memory, participants will preferentially rely upon low-level visual features because of the close semantic overlap between the encoded images and the lures (which the subjects are aware of before the experiment starts due to their experience during the practice runs; see Supplementary Fig. 11 for example image pairs); thus, we predict that recognition accuracy will correlate with lower-level reactivation during recall and that this correlation will be significantly greater than the correlation with higher-level reactivation.

Consistent with our first hypothesis, FSIC reveals neural reactivation of low-level, mid-level, high-level, and semantic features during recall throughout the cortex, including much of the dorsal and ventral visual streams, as well as the frontal cortex. As for our behavioral hypotheses, reactivation of lower- and higher-level features correlate with subjective vividness, but, contrary to our expectation, the correlation for the feature levels was approximately equivalent. Moreover, while subjects with greater lower-level reactivation within the early visual cortex during recall have higher recognition accuracy, trial-by-trial variation in low-level feature reactivation predicts correct responses only for participants with higher-than-average recognition performance on lure trials.

## Results

**Neural reactivation.** To measure neural reactivation during memory recall, an encoding-decoding approach was used[6] to

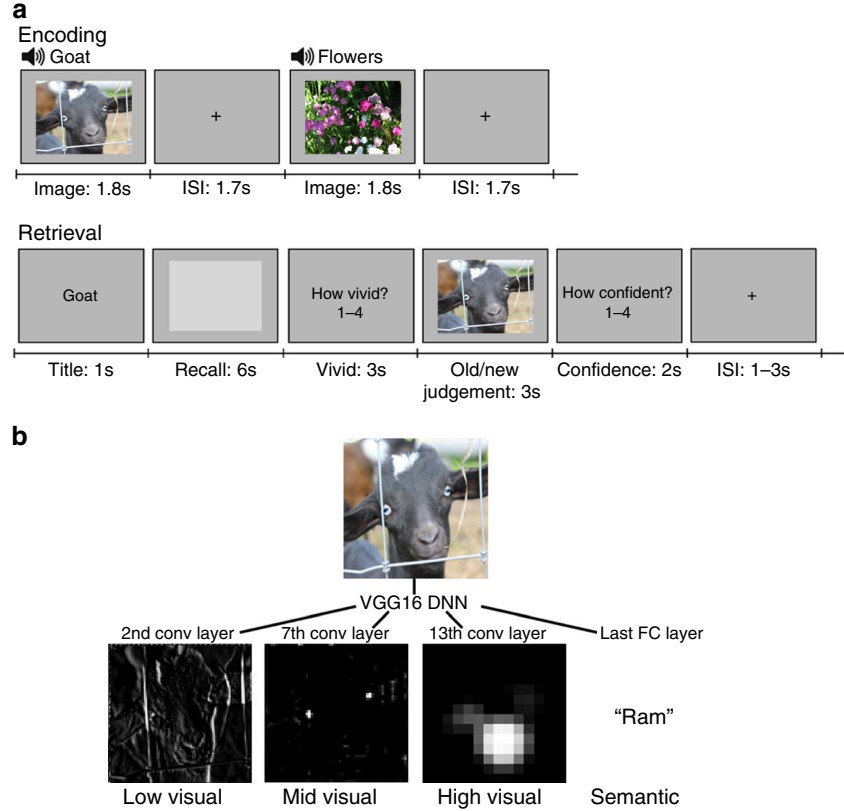

**Fig. 1 Procedure and visual features. a** Alternating image encoding and retrieval tasks. During encoding, participants performed a 1-back task while viewing a sequence of color photographs accompanied by matching auditory labels. During retrieval, participants (1) were cued with a visually presented label, (2) retrieved and maintained a mental image of the associated photograph over a 6 s delay, (3) indicated the vividness of their mental image using 1–4 scale, (4) decided whether a probe image matched the cued item, and (5) entered their confidence rating with respect to the old/new judgment. **b** For each image, features were extracted from layer node activations using the VGG16 deep neural net (DNN). Activations from the 2nd, 7th, and 13th convolutional (conv) layers, and the last fully connected layer were used, corresponding to low-visual, middle-visual, high-visual, and semantic (visual object semantics) features, respectively. Owing to copyright concerns, images used in the study could not be included in the figure. The images depicted in the figure are for explanatory purposes only.

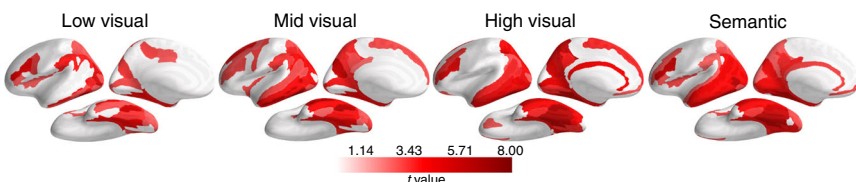

**Fig. 2 Neural reactivation during episodic recall.** Reactivation for each bilateral ROI and feature level (column = feature). Reactivation was significantly greater than chance throughout the dorsal and ventral visual streams and within the lateral and orbital frontal cortex during recall. The *t*-values are thresholded at *p* < 0.05, one-tailed bootstrap, FDR corrected.

predict the neural activity in response to a set of features comprising a seen or imagined image. The correlation between model predictions and the activity measured during visual recall was then used to decode the cued image.

Brain activity measured during the 1-back task runs and the first two video runs was used to train cortical surface-based vertex-wise encoding models for each of four visual feature levels: low-level visual features, mid-level visual features, high-level visual features, and semantic features. Given recent work showing a correspondence between visual features derived from an image recognition CNN and the features underlying human vision[57,64], the encoding models used features extracted from layer activations in a DNN (layers 2, 7, 13, and 16 of VGG16[65]) to predict neural activity (Fig. 1b).

To identify brain regions that were well-modeled by the vertex-wise feature-specific encoding models, neural activity predicted by the encoding models for each trial and feature-level were grouped into 148 bilateral cortical Freesurfer ROIs[66]. For each ROI and trial, predictions of neural activity for all encoded images were correlated with the observed neural activity during the 6-s recall period. The predictions were then sorted by correlation coefficient, and the rank of the prediction associated with the cued image was recorded. To make the rank measure more interpretable it was mean-centered so that a value significantly >0 indicates neural reactivation.

Figure 2 depicts neural reactivation for all cortical ROIs during episodic recall (for decoding performance shown time-point by time-point over the entire retrieval period see Supplementary Fig. 2). Consistent with previous findings[4,5,7,11,64], the ability to decode recalled memories was greatest throughout the dorsal and ventral visual streams for all feature levels. Significant decoding was also seen in the lateral prefrontal cortex, particularly within

the inferior frontal sulcus. Moreover, decoding accuracy for low-level features in the calcarine sulcus during perception of the recognition probe was significantly greater when using 3 by 3 features (mean rank = 11.7) vs. the same approach using 1 by 1 features (mean rank = 10.1) [$t(26)=5.51$, $p < 0.001$, two-tailed paired-samples $t$-test], indicating that some spatial representational structure was preserved despite eye movements. Overall, our findings indicate widespread neural reactivation associated with all feature-levels during episodic recall.

**Feature-specific informational connectivity.** Despite strong findings indicating reinstatement of all CNN feature-levels throughout the cerebral cortex, correlations between features from different network layers (Supplementary Fig. 3) makes it difficult to independently assess the contribution of each feature level to memory reactivation. Thus, to assess the independent contribution of each feature level to reactivation, one must statistically account for neural activity associated with all non-target features. To that end, we developed feature-specific informational connectivity (FSIC)—a variant of informational connectivity[61].

The key insight underlying FSIC is that trial-by-trial memory fidelity will naturally vary between feature levels as a result of differences in the proportion of recalled features and the extent to which the features can be used to separate the target image from the other recalled images. Each feature layer will therefore be associated with a unique (but not independent) pattern of reactivation over trials. Moreover, assuming feature-specific information is shared across regions, as suggested by the theorized role of the frontal cortex in the coordination of reactivation during recall[29–34], regions that represent the same, or very similar, feature-specific information should display similar trial-by-trial reactivation patterns.

FSIC works by extracting the trial-by-trial reactivation pattern for a given feature-level from a representative seed region and looking for a significant match in a target ROI elsewhere in the cortex. Owing to the correlation between features from different levels of the neural network (VGG16), as well as the expected

trial-by-trial correlation in the number of features recalled across feature levels (e.g., the detailed recall of low-level features may often be accompanied by the detailed recall of high-level features), we controlled for the trial-by-trial variability associated with non-target feature levels. FSIC therefore measures the correlation between trial-by-trial feature-specific neural reactivation in a seed ROI and a target ROI while regressing out the reactivation associated with all non-target feature-levels (features extracted from VGG16; see Supplementary Note 3) in the target ROI. By capturing this interregional trial-by-trial variance in reactivation fidelity, FSIC not only has greater specificity than simply assessing mean decoding accuracy, it potentially has greater sensitivity (see Supplementary Note 4).

Before applying FSIC to experimental data we validated the approach with a simulation to determine whether FSIC detects neural reactivation associated with a specific visual feature-level, while eliminating false positives. Functional magnetic resonance imaging (fMRI) data was simulated for 200 subjects using the node activations/outputs from the CNN in response to the experimental stimuli (see fMRI Data Simulation for details). Figure 3a depicts the classification accuracy results for this simulated data. Despite each ROI representing features from only one feature-level, significant effects are present for all feature-levels within each ROI. Figure 3b depicts neural reactivation results using FSIC assuming identical trial-by-trial reactivation fidelity (i.e., the proportion of recalled features) across feature-levels. In contrast to the naive classification accuracy method, FSIC accurately identifies neural reactivation associated with only the features present within each ROI, albeit with a small amount of signal smearing to features in adjacent layers. No signal smearing was found when trial-by-trial reactivation fidelity was assumed to be independent across feature-levels (see diagonal of Supplementary Fig. 4c)—an assumption that more accurately modeled the off-diagonal of Fig. 4b (compare Supplementary Fig. 4b, c)—so the simulation's results depicted in Fig. 3b likely overestimate the amount of signal smearing one can expect when applying the technique to real data. Moreover, similar, yet generally weaker, results were found when the seed ROIs

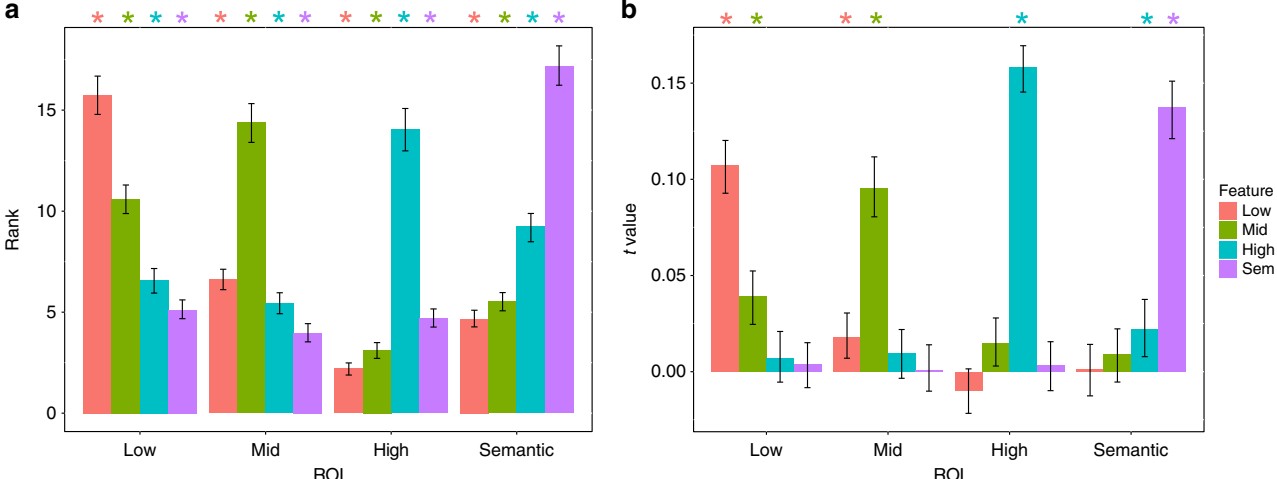

**Fig. 3 Simulated results for decoding accuracy and feature-specific informational connectivity.** fMRI data was simulated (200 simulated subjects; see Methods section) and then run through the processing pipeline for FSIC (see Methods section) to validate the approach. ROIs only contain features from the indicated feature-level. **a** Image classification performance (rank measure) for all combinations of ROI and feature-level. Correlations between feature-levels result in the classification accuracy measure falsely indicating the presence of features that are not present within the target ROI. **b** FSIC results for all combinations of ROI and feature-level assuming identical trial-by-trial memory accuracy across feature-levels. A separate seed was used for each feature-level corresponding to that feature-level (the results are also depicted in Supplementary Fig. 4b along the diagonal). Significant FSIC results only indicate the presence of the feature-level contained within each ROI, except for relatively weak evidence for the presence of adjacent feature-levels (e.g., a significant effect associated with mid-level features was found within the low-level ROI). Error bars are 90% CIs; *indicates $p < 0.05$, one-tailed bootstrap, FDR corrected.

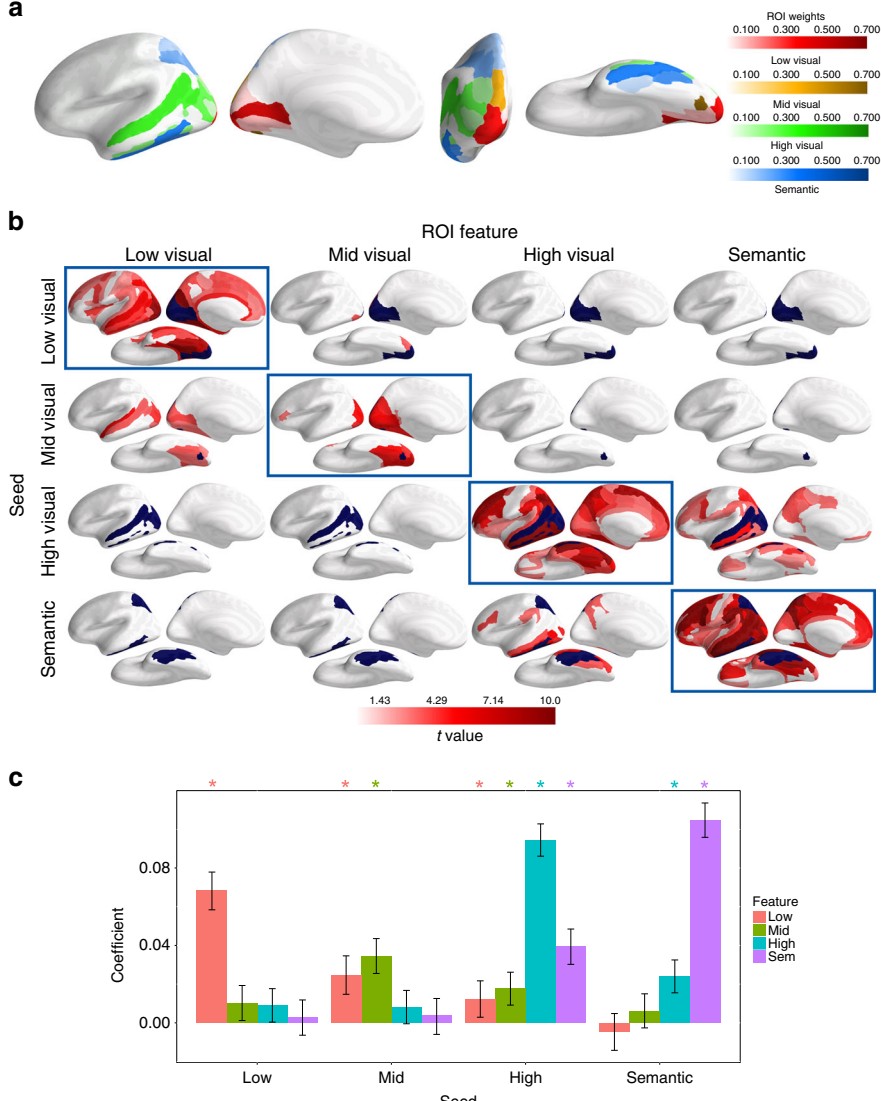

**Fig. 4 Seed ROI weights and feature-specific informational connectivity during episodic recall.** All ROIs are bilateral. **a** Seed ROI weights for each feature level. Seed ROI weights are proportional to the decoding accuracy for the target feature level relative to the other feature levels during perception of the recognition probe (i.e., the relative accuracy peaks). **b** FSIC results for all combinations of seed ROI/feature-level and target ROI feature-level. For FSIC, neural reactivation (during memory recall) of each feature level within the corresponding seed ROI (rows; seed ROIs colored dark blue) was correlated with reactivation of all four feature levels (columns), controlling for the reactivation of the non-target feature levels, within all ROIs except for the seed. The $t$-values associated with those correlations are indicated with shades of red and thresholded at $p < 0.05$, one-tailed bootstrap, FDR corrected. **c** The partial correlation coefficients from **b** averaged over all ROIs (note: this graph is not conceptually the same as Fig. 3b—this is not possible because we do not know a priori what features are represented in the ROIs). Error bars are 90% CIs; *indicates $p < 0.05$, one-tailed bootstrap, FDR corrected.

contained an equal proportion of vertices representing each feature-level (seeds in the above results only contained the target feature-level), suggesting that the feature-specificity of FSIC is not dependent on the selection of seed ROIs that only contain the target feature-level (Supplementary Fig. 4a). FSIC may therefore be used to greatly improve our ability to isolate neural reactivation associated with specific features when compared to the naive approach.

Figure 4b depicts the results obtained from applying FSIC to fMRI data measured during visual episodic recall (the results are robust to layer selection: Supplementary Fig. 5; for FSIC during recognition see Supplementary Fig. 6). The figure displays the partial correlation of neural reactivation for each feature level within the corresponding seed ROIs (rows; ROIs from Fig. 4a marked with blue; see ROI/Seed Selection in the Methods for details) and all four target feature levels within all other ROIs

(columns), controlling for all non-target feature levels (Fig. 4c depicts the partial regression coefficients from 4b averaged over all ROIs). Off-diagonal results indicate the partial correlation between different feature-levels, whereas on-diagonal results indicate the partial correlation within the same feature-level (Fig. 3 depicts a simulation of the latter). The partial regression coefficients within the diagonal were significantly greater than the coefficients within the off-diagonal [on-diagonal: mean = 0.075; off-diagonal: mean = 0.013; difference: mean = 0.063, 90% CI lower bound = 0.062, $p < 0.001$, one-tailed, paired-samples bootstrap]. According to our simulation results, the weak partial correlation coefficients within the off-diagonal indicate that trial-by-trial variation in memory reactivation is largely independent across feature-levels (Supplementary Fig. 4b, c), i.e., reactivation of one feature level is only weakly related to the reactivation of a different feature level. In contrast, the significantly greater partial

**Table 1 Low-level feature-specific informational connectivity during imagery within the frontal cortex.**

| Frontal ROI | $\beta$ | SE | t-values | Lower bound | Upper bound | p (FDR corrected) |
|---|---|---|---|---|---|---|
| Middle frontal sulcus | 0.083 | 0.021 | 3.90 | 0.050 | 0.117 | 0.004** |
| Superior precentral sulcus | 0.083 | 0.021 | 3.90 | 0.047 | 0.117 | 0.004** |
| Superior circular sulcus | 0.083 | 0.021 | 3.89 | 0.047 | 0.115 | 0.004** |
| Inferior precentral sulcus | 0.083 | 0.021 | 3.87 | 0.046 | 0.118 | 0.004** |
| Superior frontal gyrus | 0.077 | 0.022 | 3.47 | 0.039 | 0.114 | 0.004** |
| Anterior midcingulate | 0.068 | 0.022 | 3.16 | 0.034 | 0.104 | 0.004** |
| Superior frontal sulcus | 0.067 | 0.023 | 2.97 | 0.027 | 0.107 | 0.008** |
| Middle frontal gyrus | 0.057 | 0.022 | 2.61 | 0.022 | 0.093 | 0.010* |
| Anterior cingulate | 0.055 | 0.022 | 2.55 | 0.018 | 0.091 | 0.023* |
| Short insular gyri | 0.053 | 0.021 | 2.48 | 0.016 | 0.090 | 0.022* |
| Precentral gyrus | 0.053 | 0.022 | 2.44 | 0.018 | 0.089 | 0.016* |
| Inferior frontal gyrus -opercular | 0.048 | 0.022 | 2.25 | 0.013 | 0.083 | 0.020* |

The table lists the significant FSIC results (and associated statistics) within the frontal cortex depicted in the first row and first column of Fig. 4b.

regression coefficients along the diagonal indicate widespread neural reactivation for low-visual, high-visual, and semantic features that extends beyond the occipital cortex into higher-order regions of the dorsal and ventral visual streams, as well as the frontal cortex. Reactivation of mid-level features was, however, primarily limited to the occipital cortex; and this difference is not due to the relatively small size of the mid-level seed ROI (see Supplementary Fig. 7). Although expected for higher-order features[35,67–69], the widespread presence of low-level visual features within higher-order regions (see Table 1), appears to challenge a strict interpretation of the cortical visual hierarchy, which would predict results similar to what we observed for mid-level visual features.

**Relation between reactivation and vividness ratings.** With the cortical distribution of feature-specific neural reactivation established, we then assessed how feature-specific reactivation during recall relates to behavioral measures of memory performance (for the relations with reactivation during the recognition task see Supplementary Fig. 10). To test whether memory vividness (see Supplementary Note 1 for vividness behavioral results) largely results from the reactivation of lower-level visual features[22,63], measures of low- and mid-level reactivation (lower-level features), and high-level and semantic reactivation (higher-level features) were averaged together, along with the associated ROIs (Fig. 5a), forming four separate reactivation measures: one for each unique combination of feature-level and ROI. The within-subject correlations between these reactivation measures and vividness was examined with an linear mixed-effect (LME) model, where vividness rating was the dependent variable (DV), the four reactivation measures were independent variables (IVs), and the subject and image were crossed random effects (random-intercept only, due to model complexity limitations). Figure 5b shows partial regression coefficients associated with the four reactivation measures (corrected for multiple comparisons over the four coefficients using FDR). As predicted, lower- and higher-level reactivation within corresponding ROIs positively correlated with subjective vividness. Against our second prediction, however, the lower-level partial correlation coefficient was not significantly greater than the higher-level coefficient [lower-level coefficient-higher-level coefficient: 0.005, $p = 0.423$, one-tailed, paired-samples bootstrap], indicating that lower and higher-level features contribute approximately equally to subjective vividness.

In addition to the positive partial correlations, we found that reactivation of higher-level features within the lower-level ROI negatively correlated with vividness. We argue (see Supplementary Note 5) that the observed negative partial correlation

between vividness and neural reactivation of higher-level features within the lower-level ROI is consistent with a predictive coding account of perception and memory recall.

**Relation between reactivation and recognition accuracy.** We hypothesized that recognition accuracy during the old/new task (see Supplementary Note 2 for the recognition task behavioral results) would selectively correlate with reactivation associated with lower-level visual features during episodic memory recall, due to the lure image being semantically similar to the original image but differing in its low-level visual features. To test this claim, the same analytical approach described above for the correlation between reactivation and vividness was used, replacing vividness with accuracy as the DV (correct = 1, incorrect = 0). No significant correlations were found [low feature, low ROI: $\beta = 0.001$, $p = 0.972$; low feature, high ROI: $\beta = 0.037$, $p = 0.318$; high feature, low ROI: $\beta = -0.010$, $p = 0.863$; high feature, high ROI: $\beta = -0.031$, $p = 0.318$; two-tailed bootstrap, FDR corrected]. Next, we examined the correlation between-subjects using a similar model to the one used for the within-subject analysis (except the DV and IVs were within-subject averages, and subject and image were not included as random effects). As predicted, we found a significant partial correlation between recognition memory accuracy and lower-level reactivation within the lower-level ROI, which was significantly greater than the correlation with higher-level features in the higher-level ROI [lower-level coefficient - higher-level coefficient: 1.199, $p = 0.032$; one-tailed bootstrap, paired samples] (Fig. 5c; within- and between-subject coefficient $p$-values were grouped together when controlling for multiple comparisons using FDR to account for the lack of within-subject findings; see Supplementary Fig. 9a, b for the results divided into old and lure trial accuracy).

The between-subject correlation between recognition accuracy and low-level reactivation suggests that only some subjects successfully use neural reactivation within early visual regions to improve recognition memory. The null within-subject correlation between recognition accuracy and low-level reactivation might stem from this individual difference. To explore this possibility, the relation between memory accuracy and neural reactivation was calculated for each subject (using the within-subject linear model, except subject and image were not used as random effects). The resulting partial regression coefficients for each combination of ROI and feature-level were then separately correlated with the subjects' memory accuracy for all trials, lure trials, and "old" trials (Supplementary Fig. 11a–c, respectively). Significant positive correlations with lure-trial accuracy were found for lower- and higher-level features within the

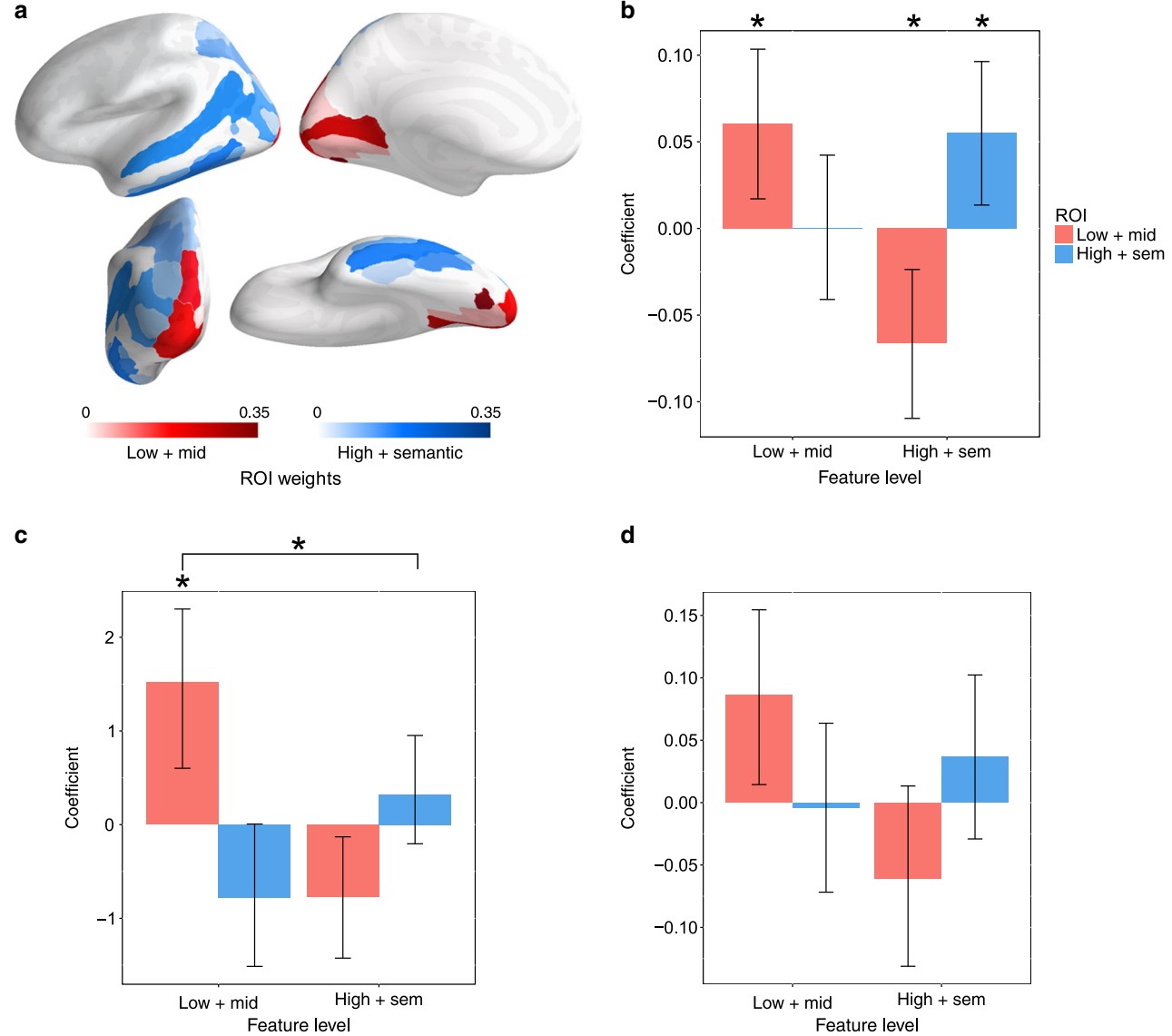

**Fig. 5 Correlations between feature-specific neural reactivation, vividness, and recognition accuracy. a** ROI weights combining the low- and mid-level and high- and semantic-level ROIs. **b** Within-subject partial regression coefficients measuring the relation between neural reactivation during recall and vividness for all combinations of feature-level and ROI. **c** Between-subject partial regression coefficients measuring the relation between neural reactivation and recognition accuracy (during the Old/New task) for all combinations of feature-level and ROI. **d** Within-subject partial regression coefficients measuring the relation between neural reactivation and recognition accuracy for the 13 subjects with the highest average "new"/lure trial accuracy. The error bars are 95% CIs; *indicates $p < 0.05$, two-tailed bootstrap, FDR corrected over the four coefficients.

corresponding ROIs, suggesting that the hypothesized positive within-subject correlation between memory accuracy and neural reactivation may only be evident for subjects with relatively high recognition accuracy. This possibility was tested using the same model as the original within-subject analysis on the thirteen subjects (half of the 27 rounded down) with the highest average memory accuracy on the lure trials (Fig. 5d; see Supplementary Fig. 8e for the 13 subjects with the lowest average memory accuracy, and see Supplementary Fig. 8c–f for the results divided into old and lure trials). For this high-performance group, we focused on the partial regression coefficients that were significant in the above analysis, i.e., the lower- and higher-level visual features within the corresponding ROIs. Of the two coefficients, only the one associated with lower-level features was significantly greater than zero [lower-level: $\beta = 0.081$, $p = 0.028$; higher-level: $\beta = 0.031$, $p = 0.320$; one-tailed bootstrap, FDR corrected], but it was not significantly greater than the higher-level coefficient

[lower-level coefficient - higher-level coefficient: 0.050, $p = 0.155$; one-tailed, paired-samples bootstrap]. Thus, there is a relationship between low-level feature reactivation and recognition memory performance, but it is limited to the higher performing subset of participants.

## Discussion

The primary goal of the current study was to reveal the feature-level composition of neural reactivation patterns measured throughout the neocortical mantle during a task requiring vivid recall of a diverse set of naturalistic images. Using FSIC, we found that visual features from all selected levels of the CNN were represented throughout the cortical visual hierarchy; but these representations were not evenly distributed across ROIs (see the diagonal of Fig. 4b). Consistent with previous work indicating a correspondence between the hierarchical organization of the

layers of a CNN and the cortical regions of the visual processing stream[57,64], the distribution of features, revealed by FSIC and the locations of peak neural reactivation for each feature-level (Fig. 4a), was organized according to the posterior-to-anterior cortical visual hierarchy.

We also showed that lower-level visual features were represented within higher-order cortical regions, and higher-level features within lower-order regions (Fig. 4b). Unlike strictly feed-forward CNN's, the cortex comprises a complex network of both feed-forward and feed-back connections that can bypass intermediate areas, facilitating direct communication between lower- and higher-order regions[70–73], thereby enabling the maintenance, modulation and combination of features at multiple levels[74–80]. For example, the inferior frontal gyrus (IFG) has been implicated in the selective maintenance of task-relevant visual information via top-down connections with the visual cortex during working memory and mental imagery[30,31,34,77,81,82]. With FSIC we showed that the IFG contains representations of visual features from all levels of the visual hierarchy during the recall of naturalistic scenes (but not during recognition; see Supplementary Fig. 6), and that the reinstatement of these representations within the IFG is correlated with the reinstatement of the same features within the occipital cortex, suggesting that the region facilitates feature-specific neural reactivation in early visual areas.

Low-level visual, high-level visual, and semantic features, but not mid-level visual features, were identified in many higher-order visual and frontal regions beyond the IFG. While this was expected of high-level and semantic features, finding low-level features represented within the frontoparietal cortex and higher-order regions of the ventral visual stream was more surprising (although, not unprecedented: Martin et al.[83] identified one higher-order region, the perirhinal cortex, that contained both visual and conceptual representations, but the visual representations were not necessarily low-level).

This raises the question of the function of such low-level features within these putative higher-order regions—a question that recent advances within the field of computational neural networks may illuminate. Like the receptive fields of neurons within the visual cortex[84,85], the nodes of feed-forward CNNs that perform visual classification and localization tasks are organized such that the lower-order layers have small receptive fields and weak semantics, whereas the higher-order layers have large receptive fields and strong semantics[86]. Consequently, the resolution of the semantic-sensitive layers is low, resulting in the loss of fine details essential for some tasks (e.g., the classification of small objects). To address this problem, recent CNNs have incorporated top-down connections and "skip" connections (which bypass adjacent layers) to directly combine the outputs of lower- and higher-order layers of the network, thereby increasing the effective resolution of the semantic-sensitive layers[87]. This approach has been proven to be effective for a variety of tasks requiring both accurate semantics and fine visual details, including classification and localization of small objects[88], and salient object detection (a key element of attentional processes) and boundary delineation (important for the coordination of grasping behavior, among other tasks)[89]. Given the functional roles of the higher-order ventral visual stream in visual object classification[90] and the frontoparietal cortex in attention and grasping behavior[91,92], we posit that the presence of low-level visual representations within these regions may likewise facilitate visual classification, attentional allocation and motor planning tasks specifically, and any task that requires both accurate semantics and fine visual details more generally.

We have demonstrated that features at all levels of the visual hierarchy are reactivated throughout the cortex during episodic recall. However, a caveat must be considered. It is possible that the finding of low-level features within the frontal cortex (Fig. 4b: top-left; Table 1) was due to correlations of neural activity unrelated to feature-reactivation across brain regions (i.e., noise correlations), and therefore not representative of low-level features within the frontal cortex. If this was the case, then we should attain very similar results if the same seed ROI is used, irrespective of feature-level. However, when the low-level seed ROIs are used for the mid-level FSIC analysis, little evidence for mid-level features within the frontal cortex was found (Supplementary Fig. 4c), providing strong evidence that our findings of feature-specific neural reactivation are not the result of noise correlations.

To investigate the functional contributions of feature-specific neural reactivation to memory, we tested whether vividness of memory recall positively correlates with neural reactivation—particularly of low-level visual features. Although previous research had found correlations between vividness and neural reactivation throughout early and late regions of the ventral and dorsal visual streams[16–20], the relative contributions of the reinstatement of lower- and higher-level visual features remained an open question. By measuring the reactivation of features from different levels of the visual hierarchy, as opposed to inferring feature-level based upon the location of reactivation (i.e., reverse inference[93]), we found that the reinstatement of lower- and higher-level visual features correlated with vividness to an approximately equal degree. While we did predict that vividness should correlate with reinstatement of both low- and high-level visual features, the low-level correlation was expected to be stronger based upon the assumption that the recall of visual details constituting a vivid memory is primarily dependent upon the reinstatement of low-level features[22,63].

This assumption, however, may overlook the inference of low-level features from high-level features. According to the predictive coding account of perception, visual experience results from the reciprocal exchange of bottom-up and top-down signals throughout the cortical hierarchy[94–99]. During perception, top-down connections convey predictions, which are compared against the perceptual input to generate an error signal. This signal is then propagated back up the hierarchy to update the predictions and enhance memory of the features that diverged from expectations[100,101]. We propose that during episodic memory recall, higher-level features are used to infer lower-level features, while the sparsely recalled lower level features that were not accurately predicted during perception serve to constrain this inference to be specific to the recalled episode (individually storing lower-level features that are effectively stored in the more compressed higher-level features would be inefficient). Therefore, according to a predictive coding account of visual recall, the number and accuracy of remembered visual details (i.e., memory vividness) should depend upon the reactivation of both high- and low-level features. Moreover, because participants were instructed not to rate generic imagery related to the cue as vivid, the top-down inference of low-level features that were not present in the encoded image should correlate negatively with vividness, which is what we found. Thus, the partial correlations between subjective vividness and feature-specific neural reinstatement are consistent with a predictive coding account of visual perception and memory recall.

Whereas our vividness results serve to demonstrate a connection between feature-specific neural reactivation and the subjective quality of memory, we were also interested in establishing the relationship between feature-specific neural reactivation and an objective memory measure: recognition memory accuracy. Although previous work[102] has shown that recognition accuracy is predicted by item/image-specific neural reactivation, there is no direct evidence that the finding was due to the reactivation of low-level features. The recognition memory task participants

performed in our study required access to fine-grained memory information to identify a probe image drawn from the same semantic category as old or new. Given the strong semantic overlap between the two images, higher-level semantic-like features alone would be unlikely to provide enough information to distinguish the images. Thus, we hypothesized that the recall of lower-level features would be required to perform well on the task. Overall, our results supported this hypothesis (Fig. 5c, d and Supplementary Fig. 8). We found that reactivation of lower-level features within the early visual cortex positively correlated with recognition accuracy within- and between-subjects, albeit the within-subject result only held for subjects with greater-than-average recognition accuracy on lure trials.

What might be the cause of this individual difference in the relationship between neural reactivation and recall accuracy? One possibility is that the participants differ in their reliance upon the reinstatement of higher- vs. lower-level features when comparing the presented image with the memorized image. Our original hypothesis that reactivation of lower-level features should positively correlate with recognition accuracy within-subjects assumed that all subjects would utilize lower-level representations when performing the task. Our failure to find the hypothesized within-subject effect appears to be the result of greater than expected individual variation in the ability or tendency of subjects to reactivate low-level visual features during memory retrieval. Future studies will be required to explore the cause and implications of these important individual differences.

The contributions of this study were fourfold. First, we developed FSIC, a measure of feature-specific neural reactivation that controls for the inherent correlations between hierarchically organized feature-levels without sacrificing sensitivity. Second, FSIC revealed that neural reactivation during episodic memory is more widespread than previously thought—particularly for low-level features (e.g., edges)—which we posit subserves numerous cognitive functions requiring both fine visual detail and accurate object/scene categorization. Third, we found that neural reactivation of lower-level and higher-level visual features contributed equally to the subjective vividness of recall, which we argue supports a predictive coding account of perception and recall. Lastly, we confirmed that reactivation of low-level visual features correlates with recognition accuracy on a task requiring fine-grained memory discrimination. Overall, the current study's results show the potential for FSIC, and other feature-specific approaches that can decompose neural pattern representations, to test and elucidate the mechanisms underpinning long held theories about the brain basis of memory and cognition.

## Methods

**Participants**. Thirty-seven right-handed young adults with normal or corrected-to-normal vision and no history of neurological or psychiatric disease were recruited through the Baycrest subject pool, tested and paid for their participation. Informed consent was obtained, and the experimental protocol was approved by the Rotman Research Institute's Ethics Board. Subjects were either native or fluent English speakers and had no contraindications for MRI. Data from ten of these participants were excluded from the final analyses for the following reasons: excessive head motion (5; removed if > 5 mm within run maximum displacement in head motion), fell asleep (2), did not complete experiment (3). Thus, 27 participants were included in the final analysis (15 males and 12 females, 20–32-year-old, mean age = 25).

**Stimuli**. One-hundred and eleven colored photographs (800 by 600) were gathered from online sources. For each image, an image pair was acquired using Google's similar image search function, for a total of 111 image pairs (222 images). Twenty-one image pairs were used for practice, and the remaining 90 were used during the in-scan encoding and retrieval tasks (see Supplementary Fig. 11 for example image pairs). Each image was paired with a short descriptive audio title in a synthesized female voice (https://neospeech.com; voice: Kate) during encoding runs; this title served as a visually presented retrieval cue during the in-scan retrieval task. Two videos used for model training (720 by 480 pixels; 30 fps; 10 m 25 s and 10 m 35 s

in length) comprised a series of short (~4 s) clips drawn from YouTube and Vimeo, containing a wide variety of themes (e.g., still photos of bugs, people performing manual tasks, animated text, etc.). One additional video cut from "Indiana Jones: Raiders of the Lost Ark" (1024 by 435 pixels; 10 m 6 s in length) was displayed while in the scanner, but the associated data was not used in this experiment because the aspect ratio (widescreen) did not match the images.

**Procedure**. Before undergoing MRI, participants were trained on a practice version of the task incorporating 21 practice image pairs. Inside the MRI scanner, participants completed three Video viewing runs and three encoding-retrieval sets. The order of the runs was as follows: first Video viewing run (short clips 1), second Video viewing run (short clips 2), third Video viewing run (Indiana Jones clip), first encoding-retrieval set, second encoding-retrieval set, third encoding-retrieval set. A high-resolution structural scan was acquired between the second and third encoding-retrieval sets, providing a break.

Video viewing runs were 10 m 57 s long. For each run, participants were instructed to pay attention while the video (with audio) played within the center of the screen. The order of the videos was the same for all participants.

Encoding-retrieval sets were composed of one encoding run followed by one retrieval run. Each set required the participants to first memorize and then recall 30 images drawn from 30 image pairs. The image pairs within each set were selected randomly, with the constraint that no image pair could be used in more than one set. The image selected from each image pair to be presented during encoding was counterbalanced across subjects. This experimental procedure was designed to limit the concurrent memory load to 30 images for each of three consecutive pairs of encoding-retrieval runs.

Encoding runs were 6 m 24 s long. Each run started with 10 s during which instructions were displayed on-screen. Trials began with the appearance of an image in the center of the screen (1.8 s), accompanied by a simultaneous descriptive audio cue (e.g., a picture depicting toddlers would be coupled with the spoken word "toddlers"). Images occupied 800 by 600 pixels of a 1024 by 768 pixel screen. Between trials, a crosshair appeared centrally (font size = 50) for 1.7 s. Participants were instructed to pay attention to each image and to encode as many details as possible so that they could visualize the images as precisely as possible during the imagery task. The participants also performed a 1-back task requiring the participants to press "1" if the displayed image was the same as the preceding image, and "2" otherwise. Within each run, stimuli for the 1-back task were randomly sampled with the following constraints: (1) each image was repeated exactly four times in the run (120 trials per run; 360 for the entire session), (2) there was only one immediate repetition per image, and (3) the other two repetitions were at least four items apart in the 1-back sequence.

Retrieval runs were 9 m 32 s long. Each run started with 10 s during which instructions were displayed on-screen. Thirty images were then cued once each (the order was randomized), for a total of 30 trials per run (90 for the entire scan). Trials began with an image title appeared in the center of the screen for 1 s (font = Courier New, font size = 30). After 1 s, the title was replaced by an empty rectangular box shown in the center of the screen (6 s), and whose edges corresponded to the edges of the stimulus images (800 by 600 pixels). Participants were instructed to visualize the image that corresponded to the title as accurately as they could within the confines of the box. Once the box disappeared, participants were prompted to rate the subjective vividness (defined as the relative number of recalled features specific to the cued image presented during encoding) of their mental image on a 1–4 scale (1 = a very small number of visual details were recalled, 4 = a very large number of visual details were recalled) (3 s) using a four-button fiber optic response box (right hand; 1 = right index finger; 4 = right little finger). This was followed by the appearance of a probe image (800 by 600 pixels) in the center of the screen (3 s), that was either the same as or similar to the trial's cued image (i.e., either the image shown during encoding or its pair). While the image remained on the screen, the participants were instructed to respond with "1" if they thought that the image was the one seen during encoding (old), or "2" if the image was new (responses made using the response box). Following the disappearance of the image, participants were prompted to rate their confidence in their old/new response on a 1–4 scale (2 s) using the response box. Between each trial, a crosshair (font size = 50) appeared in the center of the screen for either 1, 2, or 3 s.

Randomization sequences were generated such that both images within each image pair (image A and B) were presented equally often during the encoding runs across subjects. During retrieval runs each image appeared equally often as a matching (encode A -> probe A) or mismatching (encode A -> probe B) image across subjects. Owing to the need to remove several subjects from the analyses, stimulus versions were approximately balanced over subjects.

**Setup and data acquisition**. Participants were scanned with a 3.0-T Siemens MAGNETOM Trio MRI scanner using a 32-channel head coil system. Functional images were acquired using a multiband Echo-planar imaging (EPI) sequence sensitive to BOLD contrast (22 × 22 cm field of view with a 110 × 110 matrix size, resulting in an in-plane resolution of 2 × 2 mm for each of 63 2-mm axial slices; repetition time = 1.77 s; echo time = 30 ms; flip angle = 62 degrees). A high-resolution whole-brain magnetization prepared rapid gradient echo (MP-RAGE)

3-D T1-weighted scan (160 slices of 1 mm thickness, 19.2 × 25.6 cm field of view) was also acquired for anatomical localization.

The experiment was programmed with the E-Prime 2.0.10.353 software (Psychology Software Tools, Pittsburgh, PA). Visual stimuli were projected onto a screen behind the scanner made visible to the participant through a mirror mounted on the head coil.

**fMRI preprocessing.** Functional images were converted into NIFTI-1 format, motion-corrected and realigned to the average image of the first run with AFNI's (Cox 1996) *3dvolreg* program. The maximum displacement for each EPI image relative to the reference image was recorded and assessed for head motion. The average EPI image was then co-registered to the high-resolution T1-weighted MP-RAGE structural using the AFNI program align_epi_anat.py[103].

The functional data for each experimental task (Video viewing, 1-back encoding task, retrieval task) was then projected to a subject-specific cortical surface generated by Freesurfer 5.3[104]. The target surface was a spherically normalized mesh with 32,000 vertices that was standardized using the resampling procedure implemented in the AFNI program *MapIcosahedron*[105]. To project volumetric imaging data to the cortical surface we used the AFNI program *3dVol2Surf* with the "average" mapping algorithm, which approximates the value at each surface vertex as the average value among the set of voxels that intersect a line along the surface normal connecting the white matter and pial surfaces.

The three video scans (experimental runs 1-3), because they involved a continuous stimulation paradigm, were directly mapped to the surface without any pre-processing to the cortical surface. The three retrieval scans (runs 5, 7, 9) were first divided into a sequence of experimental trials with each trial beginning ($t = -2$) 2 s prior to the onset of the retrieval cue (verbal label) and ending 32 s later in 2 s increments. These trials were then concatenated in time to form a series of 90 trial-specific time-series, each of which consisted of 16 samples. The resulting trial-wise data blocks were then projected onto the cortical surface. To facilitate separate analyses of the "imagery" and "old/new judgment" retrieval data, a regression approach was implemented. For each trial, the expected hemodynamic response associated with each task was generated by convolving a series of instantaneous impulses (i.e., a delta function) over the task period (10 per second; imagery: 61; old/new: 31) with the Statistical Parametric Mapping (SPM) canonical hemodynamic response. Estimates of beta coefficients for each trial and task were computed via a separate linear regression per trial (each with 16 samples: one per time-point), with vertex activity as the dependent variable, and the expected hemodynamic response values for the "recall" and "old/new judgment" tasks as independent variables. The "recall" beta coefficients were used in all subsequent neural analyses of the "imagery"/recall period (i.e., all neural analyses except for FSIC during the recognition period) and the "old/new judgment" beta coefficients were used in all subsequent neural analyses of the "old/new judgment"/recognition period (i.e., FSIC during the recognition period; see Supplementary Fig. 6). Data from the three video scans (runs 4, 6, 8) were first processed in volumetric space using a trial-wise regression approach, where the onset of each image stimulus was modeled with a separate regressor formed from a convolution of the instantaneous impulse with the SPM canonical hemodynamic response. Estimates of trial-wise beta coefficients were then computed using the "least squares sum"[106] regularized regression approach as implemented in the AFNI program *3dLSS*. The 360 (30 unique images per run, 4 repetitions per run, 3 total runs) estimated beta coefficients were then projected onto the cortical surface with *3dVol2Surf*.

**Deep-neural network image features.** We used the pretrained TensorFlow implementation of the VGG16 deep-neural network (DNN) model[65] (see http://www.cs.toronto.edu/~frossard/post/vgg16 for the implementation used). Like AlexNet (the network used in previous studies[57]), VGG16 uses Fukushima's[107] original visual-cortex inspired architecture, but with greatly improved top-5 (out of 1000) classification accuracy (AlexNet: 83%, VGG16: 93%). The network's accuracy was particularly important for this study because we did not hand-select stimuli (images and video frames) that were correctly classified by the net. The VGG16 model consists of a total of thirteen convolutional layers and three fully connected layers. Ninety image pairs from the memory task and 3775 video frames (three frames per second; taken from the two short-clip videos; video 1: 1875 frames; video 2: 1900 frames; extracted using "Free Video to JPG Converter" https://www.dvdvideosoft.com/products/dvd/Free-Video-to-JPG-Converter.htm) were resized to 224 × 224 pixels to compute outputs of the VGG16 model for each image/frame. The outputs from the units in the second convolutional layer (layer 2), the seventh convolutional layer (layer 7), the last convolutional layer (layer 13), and the final fully connected layer (layer 16) were treated as vectors corresponding to low-level visual features, mid-level visual features, high-level visual features and semantic features, respectively.

Convolutional layers were selected to represent visual features because they are modeled after the structure of the visual cortex[107], and previous work showed that the features contained within the convolutional layers of AlexNet (which has a similar architecture to VGG16) corresponded to the features represented throughout the visual cortex[57]. The first (1), middle (7), and last (13) convolutional layers were initially selected to represent the low-, mid-, and high-level features. The layer activations were then visually inspected to confirm whether they represent the appropriate features. The low-level layer was required to have similar outputs to edge filters. Layer 2 better approximated edge filters than layer 1, so layer 2 was used instead. The high-level layer was required to have features that selectively respond to complex objects. Layer 13 contained such features (e.g., the face-selective feature in Fig. 1b), so it was retained. There were no a priori demands on the type of features represented by the middle layer, so layer 7 was retained.

The training clips/images did not contain all the object categories of the 90 images used in the encoding/retrieval parts, and some images/clips contained objects that were not in the list of 1000 ImageNet object categories. Consequently, some relevant semantic features may not be effectively mapped onto brain activity. To address these issues, VGG16's softmax output layer (the last layer of the CNN) was chosen to represent visual object semantics because it contains the probability distribution that the input image belongs to each of the 1000 pretrained ImageNet categories, thereby representing categorical confusion. Because related object categories are confused with each other in deep CNNs (e.g., "grille" confused with "convertible"; for more examples see Fig. 4 in Krizhevsky et al.[108]), the inclusion of these categorical errors reduces the sparsity of the semantic feature vector, while capturing broader (less exact) object semantics. This enables semantic feature-brain mappings to be learned by the encoding models when the training set contains images that are semantically related to the test set images, and when the training/test set images contain objects from categories that are semantically related to one or more ImageNet categories—as opposed to images from identical semantic categories (according to twelve independent raters, there are strong semantic relationships between training/test set images and the categorical labels VGG16 assigns: an average of 60% of the training/test set images had at least one label in the top 5 (out of 1000) that had a clear/direct semantic relation to the image— which was significantly greater than the 6% attained with shuffled labels ($t(10) = 12.7$, $p < 0.001$, one-tailed); see Supplementary Fig. 12 for more details). However, networks can also confuse visually similar, but semantically unrelated categories, increasing the likelihood that semantic and (high-level) visual features will be conflated. This potential confound is addressed by controlling for the correlations between feature levels—a focus of the current study.

To account for the low retinotopic spatial resolution resulting from participants eye movements, the spatial resolution of the convolutional layers (the fully connected layer has no explicit spatial representation) was reduced to 3 by 3 (original resolution for layer 2: 224 by 224; layer 7: 56 by 56; layer 13: 14 by 14). The resultant vector length of low-level visual features, mid-level visual features, high-level visual features and semantic features was 576, 2304, 4608, and 1000, respectively. Convolutional layer activations were log-transformed to improve prediction accuracy[6].

**Encoding model.** Separate encoding models were estimated for all combinations of subject, feature level and brain surface vertex[6]. Let $v_{it}$ be the signal from vertex $i$ during trial $t$. The encoding model for this vertex for a given feature level, $l$, is:

$$v_{it} = \mathbf{h}^{\mathrm{T}}\mathbf{f}_{lt} + \epsilon$$

Here $\mathbf{f}_{lt}$ is a $100 \times 1$ vector of 100 image features from the layer of VGG16 representing the target feature level, $l$, associated with the current trial/image, $t$ (only the 100 features from layer $l$ with the largest positive correlations with the vertex activity, $v_i$, were selected to make the computation tractable. Correlations were performed immediately before each non-negative lasso regression using data from the movie and encoding tasks), $\mathbf{h}$ is a $100 \times 1$ vector of model parameters that indicate the vertex's sensitivity to a particular feature (the superscript $T$ indicates transposition) and $\epsilon$ is zero-mean Gaussian additive noise.

The model parameters $\mathbf{h}$ were fit using non-negative lasso regression (R package "nnlasso"[109]) trained on data drawn from the encoding and movie viewing tasks (excluding the Indiana Jones video because its widescreen aspect ratio differed significantly from the encoded images) using threefold cross validation over the encoding data (cross validation was performed over images, so trials containing presentations of the to-be-predicted image were not included in the training set; all movie data was used in each fold). The non-negative constraint was included to reduce the possibility that a complex linear combination of low-level features may approximate one or more high-level features. The regularization parameter (lambda) was determined by testing five log-spaced values from ~1/10,000 to 1 (using the nnlasso function's path feature). For each value of the regularization parameter, the model parameters $\mathbf{h}$ were estimated for each vertex and then prediction accuracy (sum of squared errors; SSE) was measured on the held-out encoding data. For each vertex, the regularization parameter (lambda) that produced the highest prediction accuracy was retained for image decoding during recall.

**Image decoding.** Encoding models were used to predict neural activity during recall for each combination of subject, feature-level, ROI, and retrieval trial (74 bilateral cortical FreeSurfer ROIs). The accuracy of this prediction was assessed as follows: (1) for each combination of subject, feature-level, and ROI the predicted neural activation patterns for the 90 images viewed during the encoding task were generated using a model that was trained on the movie and encoding task data, excluding data from encoding trials wherein the predicted image was viewed using threefold cross validation; (2) for each retrieval trial, the predictions were correlated (Pearson correlation across vertices within the given ROI) with the observed neural activity during recall resulting in 90 correlation coefficients. (3) the

correlation coefficients were ranked in descending order, and the rank of the prediction associated with the recalled image was recorded (1 = highest accuracy, 90 = lowest accuracy). (4) This rank was then subtracted from the mean rank (45.5) so that 0 was chance, and a positive value indicated greater-than-chance accuracy for the given trial (44.5 = highest accuracy, –44.5 = lowest accuracy).

**Seed ROI selection.** The goal of the ROI seed selection was to identify the ROIs with the greatest reactivation for the target feature level relative to the non-target feature levels, controlling for mean reactivation across ROIs. The procedure for generating weight values for each ROI (Fig. 3a) was as follows: (1) compute the average classification accuracy across subjects during image perception (data taken from the old/new recognition task during the retrieval blocks) for each feature-level and ROI. (2) z-score classification accuracy across ROIs for each feature level, thereby controlling for differences in mean reactivation (across ROIs) between feature levels. (3) set all values less than zero to zero, so that ROIs with z-scores less than zero (i.e., ROI's with relatively low reactivation of the target feature-level) would not be assigned a non-zero weight. (4) For each ROI and feature-level, subtract the greatest value associated with the other feature levels from the target feature's value. (5) Set all values less than zero to zero. As a result of steps 4 and 5, only those ROIs that show greater relative reactivation for the target feature level than all other feature levels will have a non-zero weight, and this weight will be proportional to the difference between the relative reactivation of the target feature level and the greatest non-target feature level. (6) Normalize the values across ROIs to sum up to one (i.e., divide each value by the sum of all values) for each feature level. (7) set values <0.05 to 0 to retain only those ROIs that were assigned a non-negligible weight (this was done so that more non-seed ROIs could be included as targets in the FSIC analysis). (8) Normalize the values across ROIs for each feature level again, because the weights will no longer sum to one if any weights were set to zero in step 7.

**Feature-specific informational connectivity.** For the FSIC analyses, partial regression coefficients were calculated (using trial-by-trial reactivation data from the recall period) with separate LME models for all combinations seed ROI and target ROI. For each LME model, reactivation(rank measure) of the associated feature level for the seed ROI was the dependent variable (DV), reactivation for each of the four feature levels within the target ROI were the independent variables (IV), and participant and image were crossed random effects (random-intercept only, due to model complexity limitations). Statistical assessments were performed using bootstrap analyses ($n = 2409$ trials; trials with no vividness response were excluded), calculated with the BootMer function[110] using 1000 samples and corrected for multiple comparisons across ROIs using false discovery rate[111] (FDR).

**fMRI data simulation.** The simulation used the same experimental structure and stimuli (for training and testing the models) as the true experiment. For each simulated subject, 800 artificial vertices were created, with each vertex containing one, randomly selected, feature extracted from the CNN VGG16 as described in the "Deep-neural network image features" section. For each vertex, the feature-specific activation associated with the video frame or image presented at each time-point or trial was used to simulate the vertex's activity. Vertices were grouped into eight ROIs with 100 vertices each. There were two ROIs per feature-level (one representing the seed ROI, and the other representing the target ROI), such that features assigned to the vertices in each ROI were extracted from the assigned level. The two ROIs assigned the same level contained identical features, i.e., they were duplicates, except for the subsequent application of independent gaussian noise. For the analysis depicted in Supplementary Fig. 4b, the seed ROIs contained 25 vertices representing each of the four feature-levels (for 100 vertices total). Memory loss during recall was simulated by randomly setting a fraction of the features to zero. The same features were set to zero across ROIs representing the same feature level for a given trial, simulating cross-ROI information transfer. Trial-by-trial variation in memory accuracy was simulated by varying the fraction of feature loss over trials (randomly selected using a uniform distribution from 40 to 95%). Lastly, independent gaussian noise (mean 0, standard deviation 1) was added to all data, with the signal-to-noise ratio (SNR) varying across simulated subjects (either 15, 25, or 35%, equally distributed), to simulate all unaccounted-for variation in vertex activity, and individual variations thereof.

**Linear models and statistics.** Statistical assessment of mean neural reactivation (Figs. 2 and 3a) was performed using a separate LME model for each ROI, with neural reactivation as the DV and subject and image as crossed random effects ($n = 2409$ trials for all within-subject analyses; trials with no vividness response were excluded). Confidence intervals and p-values were calculated with bootstrap statistical analyses (1000 samples) using the BootMer function[110] and corrected for multiple comparisons across ROIs using false discovery rate[111] (FDR). For the within-subject correlations between feature-specific reactivation, vividness ratings (Fig. 5b), and recognition accuracy, LME models were used, with vividness ratings or recognition accuracy (correct vs. incorrect) as the dependent variable (DV), the four neural reactivation measures for each combination of ROI (lower-level and higher-level) and feature-level (lower-level and higher-level) as independent variables (IV), and participant and image as crossed random effects (random-intercept only, due to model complexity limitations). Confidence intervals and p-values were calculated with bootstrap statistical analyses (10,000 samples) using the BootMer function and corrected for multiple comparisons across coefficients using FDR. For the between-subject correlations between feature-specific reactivation and recognition accuracy (Fig. 5c), a single linear model was used ($n = 27$ subjects), with mean recognition accuracy as the dependent variable (DV) and the means of the four neural reactivation measures as independent variables (IV). Confidence intervals and p-values were generated with bootstrap statistical analyses (10,000 samples) and corrected for multiple comparisons using FDR across coefficients—including the four coefficients from the within-subject recognition accuracy LME (i.e., eight coefficients in total).

**Reporting summary.** Further information on research design is available in the Nature Research Reporting Summary linked to this article.

## Data availability

Data for all analyses covered in the article is available at https://github.com/MichaelBBone/FSIC-During-Episodic-Memory/releases.

## Code availability

Code for the analyses covered in Figs. 3–5 (i.e., the neural simulation, FSIC and behavioral correlations) is available at https://github.com/MichaelBBone/FSIC-During-Episodic-Memory.

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

## Acknowledgements
We thank Dirk Bernhardt-Walther for his helpful input. This work was supported by the Natural Sciences and Engineering Research Council of Canada (488937 to B.R.B.) and the Canadian Institutes of Health Research (152879 to B.R.B.).

## Author contributions
Conceptualization, M.B.B., B.R.B.; methodology, M.B.B., B.R.B., F.A.; software, M.B.B., B.R.B.; formal analysis, M.B.B.; investigation, F.A.; data curation, M.B.B., B.R.B.; writing–original draft, M.B.B.; writing–review and editing, M.B.B., B.R.B.; visualization, M.B.B.; supervision, B.R.B.

## Competing interests
The authors declare no competing interests.
