## [Peer Review File · Nature Communications]

Reviewers' comments:

Reviewer #1 (Remarks to the Author):

In their manuscript „Feature-Specific Neural Reactivation during Episodic Memory”, Bone, Ahmad and Buchsbaum describe an fMRI study on the neural mechanisms of how reinstatement of encoding-related activity supports episodic memory retrieval. Specifically, they use a novel method called “feature-specific informational connectivity (FSIC)” that combines two previously developed approaches: parametrization of stimuli in deep neural networks in combination with informational connectivity analysis. They describe reinstatement of both low- and high-level visual features across various brain regions including the frontal cortex, and show that these different features support different functions during long-term memory retrieval.

Overall, the paper addresses a very interesting and important question, namely which specific representational features (or formats) are reinstated to support memory retrieval. They focus specifically on frontal cortex, where the nature of the represented features is particularly unclear. Their method is highly innovative.

However, I do have a number of points that should be addressed. These concern the methodological approach, their finding of very broadly distributed accuracies across the brain and other issues.

Methods

1. Low-level visual representations are retinotopic. In order to prevent saccades, previous studies using encoding models flashed images for less than 200ms, while in the current study images are presented for a much longer time (1.8s). Thus retinotopic representations cannot be captured, or only to a small degree. The authors acknowledge this limitation and downsample the features to 3x3 in each layer. Could they ensure that at least this broad level of retinotopic representations was preserved despite eye movements? Why did they present the pictures for this long time period?
2. Vividness was “defined as the relative number of recalled visual details specific to the cued image presented during encoding”, on a 4-point scale. Does that mean that subjects were asked whether they remembered 1-4 details? How is a detail defined? What if participants recalled more than 4 details?
3. Preprocessing: If I understand correctly, the authors first segmented the retrieval data into overlapping trials (from -2s to +32s related to cue onset) and then concatenated the results and estimated single-trial parameters (corresponding to beta values) of task-specific hrf regressors. Is that correct? Has this approach been used before? Why is it advantageous to a classical GLM approach as implemented in SPM (using trial-specific regressors)?
4. How were the video frames selected that served to train the DNN (3 per second)? Is it crucial which exact frames were used? What was the video frame rate?
5. The authors write that “Layer selection was performed manually by inspection of filter activations (see Figure 1c for example activations).” – please elaborate. Which criteria were used? What happens if different layers are selected, i.e. are the results robust?
6. Encoding model: the authors write that “only the 100 features with the largest positive correlations with voxel activity were selected to make the computation tractable”. If I understand correctly, these features were determined during the fit to the video and encoding data? Were these features the same

for each voxel? How were they distributed across layers?

7. The rationale for the fMRI data simulation should be explained in greater detail.

8. Choice of ROIs: Using 148 ROIs is a very large number to start the analysis with. Based on previous research that only showed a correspondence between the visual pathway and vision DNNs using 148 ROIs that even expand to the frontal cortex looks a bit like a random choice that was rather done to have higher chances of finding results. Maybe a searchlight approach could have been used alternatively.

9. I would be interested to know why the authors chose to use VGG16. At one point they are referring to the Güçlü and van Gerven paper (2015) (p. 10 in the manuscript) claiming that this was the reason to settle for VGG16, though in most previous papers and especially in the van Gerven 2015 the AlexNet network was used.

10. The second problem applying VGG16 to frontal regions, but also in general, is the claim of interpreting the last fully connected layer of VGG16 as "semantic". Though previous studies could show this parallel hierarchy between visual regions and DNNs, the last layer in the network is a softmax layer. This in turn means that you are looking at percentages, which only marginally reflect what we would label to be semantic. Usually you only look at this layer if the dataset you used can be classified by the network.

11. As it seems the authors use a pre-trained version of the VGG16 network, most likely using 1000 labels from the Imagenet database. However, this database does not include stimuli from the categories that the authors used such as images with people or babies or other objects that are not among those 1000 labels. This means that the network will misclassify these images and will result in two things. On one hand you can no longer be confident in interpreting the last layer (fc16) because it will give you false "semantics". The other problem is that labels which the network is not trained on will result in strange or at least not really usable unit activations at some point in the network – most likely in the fc layers, because the already trained and tuned weights will still try to somehow classify the image input.

12. Training set: Overall the way and the number of stimuli used for training quite differ from previous studies. Other studies (e.g. Seeliger, 2018) used far more stimuli and also used a different set for training, testing and validating (in addition to the cross validation). You usually use around 1000 stimuli which are presented at least 6-8 times for training and testing and for validation around 50-100 with around 8-10 repetitions. Using video clips might be more interesting for the test subjects but you cannot compare the number of frames with the number of different images. Also the training material (videos) largely consists of content that cannot be classified by the network (people, text, see Supplement).

13. p. 29: why was the data of the third video (from the Indiana Jones clip) not used?

Results

1. How were the vividness ratings distributed across trials – i.e., did participants tend to rate similar images as highly vivid or different images? This is important because it would suggest that vividness is either driven by the stimuli (if participants strongly agree on vividness ratings) or by personal evaluations (if there is little agreement).

2. The brain-wide decoding accuracy values should be analyzed in greater detail. In particular, in which areas do the accuracies differ between the different levels? Is there indeed higher accuracy for low-level features in low-level visual areas and for high-level features in higher-level visual areas – and how are

low- and high-level visual areas defined? Is there differential accuracy in prefrontal ROIs? Shouldn't this analysis also allow to account for the correlation between the different feature levels?

3. Related to the previous point: In addition to these results, the actual accuracy values should be reported. This is particularly important since the authors trained their model on a total of only 21 minutes of visual stimulation rather than several hours of stimulus presentation in previous encoding studies.

4. Relationship to behavior: first of all, did the authors correct for multiple comparisons across feature levels and ROIs? This would be necessary.

5. The explanation in terms of predictive coding seems a bit far-fetched and very indirect. Does that mean that participants imagine two different images, one bottom-up and the other one top-down? Is there any evidence for this?

Reviewer #2 (Remarks to the Author):

Bone and colleagues report a human fMRI study that uses convolutional neural networks to measure representations of visual scenes at multiple feature levels and to then test for reactivation of these scenes, at each feature level, across multiple cortical areas. The paradigm is relatively straightforward with the key component being the retrieval phase, where subjects first recall a scene and then indicate whether a probe scene is the target vs. a lure. The paper is generally well written and addresses topics of potentially broad interest.

The first major claim is that feature information, at every level, is reactivated throughout much of cortex. This is consistent with many prior studies (including from these authors) showing that reactivation is widely distributed across many cortical areas (including frontoparietal areas). There is also an analysis that is described called feature-specific informational connectivity (FSIC). This analysis is framed as a novel method, but the idea of correlating information-based measures across brain regions is not new (see work from Coutanche, as the authors note). But even the specific application of this method to measures of memory reactivation is not new (Kuhl, Johnson, Chun 2013). It is just the specific application of this method to reactivation indexed by a CNN that has not previously been reported. So, it is a fairly incremental advance. Second, reactivation at multiple levels was associated with vividness, which again is consistent with a number of prior studies (several from these authors) that have shown that reactivation in multiple brain regions (visual cortex and beyond) is related to vividness. Third, subjects that showed stronger reactivation showed better recognition accuracy in the old/new test. The relationship between level-specific measures of reactivation and performance on the recognition memory test is potentially the most interesting aspect of the present paper, but the analyses/results are not as selective (or convincing) as they would need to be for this to support strong conclusions. Namely, the relationship between reactivation and recognition accuracy was only observed in an across subject correlation (which is a less-preferred measure than within-subject relationships) and was not selective to the specific recognition memory trials (new trials) that tested discrimination between similar memories. Collectively, the paper touches on several interesting ideas and uses very sophisticated

methods. In particular, the use of CNNs to index memory reactivation at different levels is very interesting and is likely to appeal to readers. That said, the current results ultimately represent mostly incremental advances given that there are several other studies that already report similar findings.

Major Comments

1. Figure 4b is referenced as the critical evidence that the FSIC analysis revealed feature-selective reactivation in feature-relevant cortical areas. However, there does not appear to be any direct statistical test to support this claim of selectivity. Rather, it is based on a qualitative assessment of the regions that 'showed up' in each analysis. The authors refer to the comparison of the on diagonal vs. off diagonal effects, but it does not appear that there is a direct statistical test of whether the on-diagonal effects are significantly stronger than the off diagonal effects. This analysis may well turn out a positive effect (significant difference), but I think it is important that these claims about selectivity be based on a statistical comparison.

2. Related to the above: the graph for the FSIC simulation conveys a lot more information than does the figure which shows the actual (non-simulated) results. The 'real data' figure just shows ROIs on brains and it is harder to assess the effect sizes, the comparison across ROIs/feature levels, etc.

3. The feature specific informational connectivity analysis (FSIC) is fairly confusing. It was not at all intuitive to me how/why this measure is preferable to other approaches. And I also had trouble following exactly how it was derived (ultimately, I understood it, but it took a few reas).

4. More specifically, the Results section that that the FSIC analysis covaries out all non-target feature levels. Elsewhere it describes the analysis as "controlling for all non-target feature levels". But it is not clear to me how (or why) this analysis controls for the correlation across feature levels. Was there some step taken in this analysis other than just correlating the ranks? The methods section does not provide any clarifying detail about this.

5. Also, my understanding is that FSIC approach doesn't actually test whether reactivation is "above chance" but only tracks whether two regions represent similar information (whether or not that information is accurate). Assuming this interpretation is correct, this should be made clearer and claims should be adjusted accordingly. For example, if FSIC is only tracking similarity of information across regions then, on its own, it does not establish whether or not feature level reactivation is present. Rather, two regions could each have completely inaccurate feature-level reactivation, but these 'representations' could be highly correlated across regions.

6. The data in 5b is from a within-subject analysis whereas the data in 5c is from a between subjects analysis. This switching between different kinds of measures is not ideal. And the selectivity of the relationship in 5b to the low-level features is undermined a bit by the fact that the relationship for the low-level features is likely not significantly stronger than the relationship for the high-level features

(though, this statistic is not reported).

7. For the analyses relating reactivation to recognition memory (which, again, I think is an interesting analysis), the primary/initial analyses are based on overall recognition memory (old/new) accuracy. However, it would seem that the predictions should be very different for the old vs. the new trials. The authors are testing the idea that only low-level reactivation will predict recognition accuracy with the idea that low level details are necessary to discriminate the new items (which are very similar to old items) from the old items. But subjects are not actually selecting between old and new items—rather, they are making old/new judgments EITHER about old or new items. For the old trials, it seems that reactivation at any level (high OR low) would be likely to lead subjects to (correctly) endorse the item as ‘old’ (hits). But, for the new trials, high-level reactivation of semantic information would potentially lead subjects to erroneously endorse the new items as old (false alarms) whereas low-level details should be helpful in rejecting the new item (correct rejection). Thus, it would seem that to test their critical prediction, the authors should restrict all of these analyses to the new trials. This is done for a couple of the post-hoc analyses, but it seems like this should be the key focus of all of the recognition memory analyses.

8. Also, even if the issues above are ignored, it is not clear that the relationship between low-level reactivation and recognition accuracy is STRONGER than the relationship between high-level reactivation and subsequent memory. In other words, there are some effects for low but not high vividness trials, but in order to support the prediction that low level reactivation is MORE IMPORTANT, this would require a statistical difference between the coefficients for low vs. high level reactivation.

9. The authors repeatedly emphasize the importance of feature specific representations and “decomposing” memory reactivation. There are two recent papers on these specific topics that should be discussed (or at least cited): Favila, Samide, Sweigart, & Kuhl, 2018; Lee, Samide, Richter, & Kuhl, 2018. The paper by Favila specifically considers feature-level memory reactivation across cortical areas and the relation of these representations to behavior. And the paper by Lee explicitly discusses “decomposing memory reactivation” in order to test how/whether different levels of reactivation predict accuracy in a memory test that includes targets and highly similar lures (exactly as in the current study). However, in the paper by Lee et al., analyses were separated for old and new trials (see Comment 7) and it was found that for the new items (which are the critical trials) more generic reactivation predicted false alarms whereas more detailed reactivation predicted correct rejections. The similarity of these prior papers to the current work also reduces the novelty / conceptual advance of the current study.

Minor Comments

10. Related to the analysis in 5c, I was confused by the statement that “within- and between-subject coefficients were pooled together.” I thought this analysis only used the averaged within-subject effects—in other words, how could there be within-subject coefficients if the within-subject data was

averaged together?

11. The seed ROI selection section (line 744) is very confusing. In part it is hard to follow the steps that were actually taken, but it is even less clear why these steps are being taken.

12. While I think there is reason to have some faith that the layers taken from the CNN to represent early, mid, late visual and semantic information are meaningful, it should be noted that these are not independently validated (e.g., through behavioral analyses). Put another: how do we know that these particular layers map onto psychologically-relevant early, mid, late, and semantic levels? I don't view this as a major problem, but it might be something that could be briefly touched on in the Discussion (i.e., what assumptions are being made in interpreting the CNN layers?).

13. Line 725: "For each value of the regularization parameter, the model parameters h were estimated for each voxel and then prediction accuracy (sum of squared errors; SSE) of the recognition data was measured using 3-fold cross validation. For each voxel, the model parameters h that yielded the highest prediction accuracy were retained for image decoding." I found this a bit confusing. I assume the recognition data is the old/new portion of the retrieval blocks? How exactly was the 3-fold cross validation implemented? And why was the recognition data used for this part (as opposed to the encoding trials)?

14. Perhaps I missed this somewhere, but was the significance of the reactivation measures assessed by permutation tests?

Reviewer #3 (Remarks to the Author):

Bone and colleagues investigate how activity predicted from various hierarchical layers of a deep convolutional neural network (CNN) maps onto the brain activity patterns recorded with fMRI during a memory recall task, where participants are asked to vividly imagine previously seen images. They develop a method called feature-specific informational connectivity (FSIC) that effectively regresses out features that are correlated between layers, as demonstrated with simulated data. Apart from this methodological aspect, the paper also describes a number of interesting empirical findings. Most interestingly maybe, low-level visual features of the images are not confined to low-level visual cortex during recall/imagery, but are represented throughout high-level areas including parietal and frontal lobes. They also find that the activation of both high and low-level features contributes roughly equally to the vividness of mental imagery. Somewhat less robustly, low-level feature reactivation is shown to be beneficial for discriminating a previously seen target image from a highly similar lure image.

This was a very interesting read, well written despite the complexity of the methods. The paper could make a novel contribution to the literature both from a methodological and an empirical perspective. I have a number of comments, however, that should be addressed in a revision.

Methodological comments:

(1) The pure recall/imagery part of each retrieval trial was followed relatively closely in time (7 sec) by visual presentation of a recognition stimulus (an old target or a very similar lure). The authors state in the methods section that they separately model the hemodynamic responses to recall and recognition onsets. However, modelling both contributions does not rule out that the recall regressor picks up variance shared with the recognition regressor, and there is thus a concern that the main results are partly produced by an overlap in the sensory input between training and test data, and not by pure recall/mental imagery activity. This concern could be addressed by regressing out the shared variance from the cued recall beta coefficients.

(2) Related to the first point, the authors state in the last sentence of the “Encoding models” section (l. 725-29) that the encoding models were optimized for predicting the recognition data. This statement implies that model training already involved the recognition phase data, which are naturally highly correlated with the cued recall (test) data. This could cause a major confound and should be clarified.

(3) In the selection procedure for seed ROIs, what is the reason for setting below zero values to zero in steps 3 and 5, and why was it necessary to normalize the values twice (steps 6 and 8)? These steps should be justified in the methods section.

(4) The rationale for using the videos for training is not entirely clear, and deserves a few sentences in the methods section. In particular, did these videos contain all the 90 stimulus concepts used in the later encoding/retrieval parts?

(5) Minor clarification comment: the authors should state whether they used Spearman or Pearson R for correlating predicted activity with recall activity.

Results:

(6) When reporting the behavioral data, the authors should report separately the hit rates and misses to old target items, and the correct rejections and false alarms to new similar lure images. It would also be interesting to see the relationship of feature reactivation of the different layers specifically on trials where participants made false alarms, contrasting the maps with correct old judgments. The idea being that a very generic (high-level) reactivation will be related to a higher likelihood of judging a similar lure as old (as found e.g. in Lee et al., 2018, CerebCortex). I understand though if not enough trials are available to separately investigate these conditions.

(7) It would be very helpful if a supplementary figure could be included showing the model fits for the different layers during the video/encoding runs (i.e., using the cross-validation results). Such a figure would allow the reader to directly compare layer-specific predictions when the model is tested on visual perception versus memory recall. This is particularly interesting with respect to the surprising finding that the mid-layers of the CNN map onto occipital lobe only during recall.

(8) Do the authors have information about what layers in their network are most sensitive to spatial frequency of the images? Fig. 1c suggests that spatial frequency differs between layers. Apart from the perceptual-to-semantic dimension which is the focus of the discussion in the paper, an additional detail-to-gist dimension might explain some of the differences in the main results, and is worth discussing.

(9) The result that mid-layer representations are constrained to occipital lobe is quite surprising. The authors should discuss whether there is something particular about the 7th layer e.g. in terms of spatial frequency (see previous comment). If they already have the results also for neighbouring layers 6 and 8, it would be interesting to hear whether the same result is true for those layers (but to be clear, this is not a requested re-analysis).

Conceptual:

(10) I find it somewhat misleading to call the method developed here a connectivity method, because (i) similarity/correlation between two areas does not imply connectivity, and (ii) the method effectively regresses out the overlap with other ROIs, and thus seems to amplify dissimilar patterns. This should be clarified.

(11) A number of papers should be discussed in the context of the present study, including (i) Dijkstra et al. (2018, eLIFE) suggesting it is mainly the late visual processing components that are active during later imagery; (ii) Linde-Domingo et al. (2019, Nat Comm) suggesting that higher-level semantic information is reactivated more rapidly during recall than lower-level perceptual information; (iii) Martin et al. (2018, eLIFE) suggesting that integrated low- and high-level features are represented at very late stages of the visual processing hierarchy. In addition, it might be worth discussing how FSIC relates to other recently developed information-theoretical frameworks as published in Zhan et al. (2019, Curr Biol).

Response to Reviews

We thank the reviewers for their very helpful and insightful reviews. We have tried to address all of their concerns and updated the manuscript accordingly. Our detailed point-by-point response to their remarks is below.

Reviewer #1 (Remarks to the Author):

In their manuscript „Feature-Specific Neural Reactivation during Episodic Memory”, Bone, Ahmad and Buchsbaum describe an fMRI study on the neural mechanisms of how reinstatement of encoding-related activity supports episodic memory retrieval. Specifically, they use a novel method called “feature-specific informational connectivity (FSIC)” that combines two previously developed approaches: parametrization of stimuli in deep neural networks in combination with informational connectivity analysis. They describe reinstatement of both low- and high-level visual features across various brain regions including the frontal cortex, and show that these different features support different functions during long-term memory retrieval.

Overall, the paper addresses a very interesting and important question, namely which specific representational features (or formats) are reinstated to support memory retrieval. They focus specifically on frontal cortex, where the nature of the represented features is particularly unclear. Their method is highly innovative.

However, I do have a number of points that should be addressed. These concern the methodological approach, their finding of very broadly distributed accuracies across the brain and other issues.

Methods

1. Low-level visual representations are retinotopic. In order to prevent saccades, previous studies using encoding models flashed images for less than 200ms, while in the current study images are presented for a much longer time (1.8s). Thus retinotopic representations cannot be captured, or only to a small degree. The authors acknowledge this limitation and downsample the features to 3x3 in each layer. Could they ensure that at least this broad level of retinotopic representations was preserved despite eye movements? Why did they present the pictures for this long time period?

We appreciate your detailed and insightful comments.

Yes, we find that 3x3 retinotopic representations are preserved despite eye-movements. For low-level features in the calcarine sulcus during perception of the recognition probe we found

significantly greater image classification accuracy (rank measure) using the original 3x3 features (mean rank = 11.7) vs. the same approach using 1x1 features (mean rank = 10.1) [$t(26) = 5.51$, $p < .001$, two-tailed paired-samples t-test]. Added to results (page 11, line 209).

A previous study of complex scene recall (Johansson et al., 2012) has shown that eye-movements towards the locations of salient regions of the scene take place even when eyes were centrally fixated during encoding, which suggests that people move their "mind's eye" to different regions of the image resulting in retinotopic shifts of the neural representation. Allowing subjects to shift their gaze during encoding of the images over 1.8s not only made encoding (and consequently recall) more naturalistic (which is a general focus of our lab, e.g. Bone et al. 2019), it also trained the encoding model to better account for the possibility of imagined "gaze-shifts" during retrieval.

2. Vividness was "defined as the relative number of recalled visual details specific to the cued image presented during encoding", on a 4-point scale. Does that mean that subjects were asked whether they remembered 1-4 details? How is a detail defined? What if participants recalled more than 4 details?

Vividness was defined subjectively as "the relative number of recalled visual details specific to the cued image presented during encoding", i.e. 4 = a very large number of visual details were recalled, and 1 = a very small number of visual details were recalled (clarification added to page 31, line 672). Thus, the instruction did not specify an absolute number of details for a given ordinal rating.

3. Preprocessing: If I understand correctly, the authors first segmented the retrieval data into overlapping trials (from -2s to +32s related to cue onset) and then concatenated the results and estimated single-trial parameters (corresponding to beta values) of task-specific hrf regressors. Is that correct? Has this approach been used before? Why is it advantageous to a classical GLM approach as implemented in SPM (using trial-specific regressors)?

The purpose of this approach was to first attain a time-sequence of reactivation over the entire retrieval period to verify that the signals associated with each task were clearly separated (see the new Supplementary Figure 2). We then used hemodynamic regressors on this time-sequence data to attain beta values for the "recall" and "old/new judgment" tasks. This permitted us to do a time-resolved analysis in addition to more sensitive trial-specific beta analysis using the same data. Results should be nearly identical to the common approach, because we are essentially performing the same analysis, but divided into two steps.

4. How were the video frames selected that served to train the DNN (3 per second)? Is it crucial which exact frames were used? What was the video frame rate?

Frames were extracted using “Free Video to JPG Converter” (<https://www.dvdvideosoft.com/products/dvd/Free-Video-to-JPG-Converter.htm>) (updated page 35, line 752). We selected 3-frames-per-second because this is near the limit of the temporal resolution of the hemodynamic response, and the clips (approximately 4s long) generally did not contain rapid changes in content, i.e. frames separated by 1/3s were almost always very similar. This, combined with the low spatial resolution of the features used in the study (3x3), made the exact specification of the frames used unnecessary. The video frame rate was 30-frames-per-second (added frame rate to page 29, line 626).

5. The authors write that “Layer selection was performed manually by inspection of filter activations (see Figure 1c for example activations).” – please elaborate. Which criteria were used? What happens if different layers are selected, i.e. are the results robust?

Thank you for pointing out this oversight.

Three visual features layers (low-, mid- and high-level) were selected from the convolutional layers, and one semantic layer was selected from the final layer of the network. Convolutional layers were selected to represent visual features because they are loosely modeled after the structure of the visual cortex (Fukushima, 1980), and previous work showed that the features contained within the convolutional layers of AlexNet (which has a similar architecture to VGG16) corresponded to the features represented throughout the visual cortex (Güçlü & van Gerven, 2015). The first (1), middle (7), and last (13) convolutional layers were initially selected to represent the low-, mid- and high-level features. The layer activations were then visually inspected to confirm whether they represent the appropriate features.

The low-level layer was required to have similar outputs to edge filters. Layer 2 better approximated edge filters than layer 1, so layer 2 was used instead. The high-level layer was required to have features that selectively respond to complex objects. Layer 13 contained such features (e.g. the face-selective feature in Figure 1c), so it was retained. There were no a priori demands on the type of features represented by the middle layer, so layer 7 was retained.

VGG16's softmax output layer (the last layer of the CNN) was chosen to represent visual object semantics because it contains the probability distribution that the input image belongs to each of the 1000 pretrained ImageNet categories, thereby representing categorical confusion. Because related object categories are often confused with each other in deep CNNs (e.g. "grille" confused with "convertible"; for more examples see figure 4 in Krizhevsky, Sutskever, & Hinton, 2012), the inclusion of these categorical errors reduces the sparsity of the semantic feature vector, while capturing broader (less exact) object semantics. This enables semantic feature-brain mappings to

be learned by the encoding models when the training set contains images that are semantically related to the target images, as opposed to images from an identical semantic category.

The section “Deep Neural Network Image Features” has been updated with the above information (pages 36-37, lines 759-788).

To examine the robustness of our results to the selection of CNN layers, we performed a variant of our FSIC analysis (Figure 4b) using all layers. We first subdivided the sixteen layers into four groups, with convolutional layers 1-4, 5-9, 10-13 and fully connected layers 14-16 assigned to the low-, mid-, high-visual and semantic groups, respectively. The reactivation values for each of these layers were then averaged, producing a single reactivation value for each of the four groups for each combination of subject, ROI and trial. FSIC was then performed using these values in a manner identical to the original approach. The results (Supplementary Figure 3) were very similar to the original results (Figure 4b), suggesting our approach is robust to layer selection.

Reference to Supplementary Figure 4 added to page 14, line 297.

6. Encoding model: the authors write that “only the 100 features with the largest positive correlations with voxel activity were selected to make the computation tractable”. If I understand correctly, these features were determined during the fit to the video and encoding data? Were these features the same for each voxel? How were they distributed across layers?

Thank you for pointing out that the description of the encoding model was unclear. The “Encoding Model” section has been updated (page 37, line 797-805).

“If I understand correctly, these features were determined during the fit to the video and encoding data?”

This is correct. Correlations were performed immediately before each non-negative lasso regression using the same cross validated training data.

“Were these features the same for each voxel?”

No. They were selected separately for each voxel.

“How were they distributed across layers?”

There was no distribution across layers because they were selected from each feature level/layer separately. This was done to reduce the number of features passed to the non-negative lasso regression, e.g. from ~5000 for the high-level features (layer 13) to 100.

7. The rationale for the fMRI data simulation should be explained in greater detail.

The purpose of the data simulation was to determine the feature-selectivity of FSIC. It was necessary to determine if FSIC erroneously detects features within an ROI that does not contain features from that feature level. Brain data couldn't be used for this purpose because we do not know what features are represented in a given cortical region—determining the distribution of visual features across brain regions was the primary goal of the current study. In the place of actual brain data, we simulated ROIs that contained features from only one feature level, and investigated whether FSIC (and the commonly used mean accuracy approach) were prone to finding representations of feature-levels not present in the target ROI.

“(fMRI data couldn't be used for this purpose because it was necessary to have ROIs that only represent a single feature level, and to know what that feature level is—and determining the distribution of visual features across brain regions was an empirical question taken up by this study).” Added to page 13, line 270.

8. Choice of ROIs: Using 148 ROIs is a very large number to start the analysis with. Based on previous research that only showed a correspondence between the visual pathway and vision DNNs using 148 ROIs that even expand to the frontal cortex looks a bit like a random choice that was rather done to have higher chances of finding results. Maybe a searchlight approach could have been used alternatively.

The primary goal of the study was to look at feature-level representations throughout the entire neocortex. The use of all (148) neocortical Freesurfer ROIs was appropriate for this purpose. We have previously used Freesurfer ROIs for multivoxel fMRI analyses (Arsenault and Buchsbaum, 2015; Arsenault and Buchsbaum, 2016; Bone et al. 2019; Du et al. 2016), so it is not arbitrary. Freesurfer ROIs are subject-specific and therefore are better attuned to individual anatomical features that are not captured by group-level ROI atlases. The ROI approach also has the benefit of reducing multiple comparisons relative to a searchlight approach, thereby potentially increasing sensitivity to weaker effects—like those that may have existed in the frontal cortex.

9. I would be interested to know why the authors chose to use VGG16. At one point they are referring to the Güçlü and van Gerven paper (2015) (p. 10 in the manuscript) claiming that this was the reason to settle for VGG16, though in most previous papers and especially in the van Gerven 2015 the AlexNet network was used.

We now add a clarification to this matter. “Like AlexNet (the network used in many previous studies, e.g. Güçlü & van Gerven, 2015), VGG16 uses Fukushima's (1980) original visual-cortex inspired architecture, but with greatly improved top-5 (out of 1000) classification accuracy (AlexNet: 83%, VGG16: 93%). The network's accuracy was particularly important for this study

because we did not hand-select stimuli (images and video frames) that were correctly classified by the net.” Added to page 35, line 744.

10. The second problem applying VVG16 to frontal regions, but also in general, is the claim of interpreting the last fully connected layer of VGG16 as "semantic". Though previous studies could show this parallel hierarchy between visual regions and DNNs, the last layer in the network is a softmax layer. This in turn means that you are looking at percentages, which only marginally reflect what we would label to be semantic. Usually you only look at this layer if the dataset you used can be classified by the network.

This is an insightful point. We now add the following clarifying paragraph to the manuscript:

“The training clips/images did not contain all the object categories of the 90 images used in the encoding/retrieval parts, and some images/clips contained objects that were not in the list of 1000 ImageNet object categories. Consequently, some relevant semantic features may not be effectively mapped onto brain activity. To address these issues, VGG16's softmax output layer (the last layer of the CNN) was chosen to represent visual object semantics because it contains the probability distribution that the input image belongs to each of the 1000 pretrained ImageNet categories, thereby representing categorical confusion. Because related object categories are often confused with each other in deep CNNs (e.g. "grille" confused with "convertible"; for more examples see figure 4 in Krizhevsky, Sutskever, & Hinton, 2012), the inclusion of these categorical errors reduces the sparsity of the semantic feature vector, while capturing broader (less exact) object semantics. This enables semantic feature-brain mappings to be learned by the encoding models when the training set contains images that are semantically related to the test set images, and when the training/test set images contain objects from categories that are semantically related to one or more ImageNet categories—as opposed to images from identical semantic categories. However, networks can also confuse visually similar, but semantically unrelated, categories, increasing the likelihood that semantic and (high-level) visual features will be conflated. This potential confound is addressed by controlling for the correlations between feature levels—a focus of the current study.” Added to page 36, line 771.

11. As it seems the authors use a pre-trained version of the VGG16 network, most likely using 1000 labels from the Imagenet database. However, this database does not include stimuli from the categories that the authors used such as images with people or babies or other objects that are not among those 1000 labels. This means that the network will misclassify these images and will result in two things. On one hand you can no longer be confident in interpreting the last layer (fc16) because it will give you false "semantics". The other problem is that labels which the network is not trained on will result in strange or at least not really usable unit activations at some point in the network – most likely in the fc layers, because the already trained and tuned weights will still try to somehow classify the image input.

As we discuss in the response above, these "false" semantic classifications in the softmax layer tend to capture semantically related categories or the semantics of other objects in the image. The network may also confuse visually similar categories, i.e. categories with significant overlap of high-level visual features. Consequently, the use of some images containing objects that are not within the list of 1000 ImageNet semantic categories will result in two potentially problematic outcomes: a reduction in the power of the study to detect semantic features (our strong semantic findings indicate that this is not a concern for this study), and an increased likelihood that semantic and high-level visual features will be conflated. The latter concern is addressed by controlling for the correlations between feature levels.

12. Training set: Overall the way and the number of stimuli used for training quite differ from previous studies. Other studies (e.g. Seeliger, 2018) used far more stimuli and also used a different set for training, testing and validating (in addition to the cross validation). You usually use around 1000 stimuli which are presented at least 6-8 times for training and testing and for validation around 50-100 with around 8-10 repetitions. Using video clips might be more interesting for the test subjects but you cannot compare the number of frames with the number of different images. Also the training material (videos) largely consists of content that cannot be classified by the network (people, text, see Supplement).

We used different encoding data for training and validating (separated via cross validation as described on page 38, line 811), and recall data for testing. Consequently, there was no overlap of samples used for training, validation and testing.

Each encoding model was trained on 60 images presented 4 times each and two videos containing a total of approximately 500 short clips presented once each. The relatively small amount of training data is expected to lower our per-subject power, but our overall power is increased by testing a relatively large number of subjects (27 vs. 5 (Horikawa & Kamitani, 2017), 2 (Güçlü & van Gerven, 2015), 15 (Seeliger, 2018)), and limiting our analyses to group-level effects. Using a relatively large number of subjects is preferable to greater per-subject power in the context of a memory study because significant individual differences are expected (and of interest).

"Also the training material (videos) largely consists of content that cannot be classified by the network (people, text, see Supplement)."

See the responses to comments 10 and 11.

13. p. 29: why was the data of the third video (from the Indiana Jones clip) not used?

The aspect ratio didn't match the images used during encoding/retrieval (the video was widescreen, whereas the clips/images were not), so the downsampled (3X3) feature-spaces would not have matched. We are using the Indiana Jones video in other work to examine the usefulness of hyperalignment methods for examining memory.

“...because the aspect ratio (widescreen) did not match the images.” Added to page 29, line 630.

Results

1. How were the vividness ratings distributed across trials – i.e., did participants tend to rate similar images as highly vivid or different images? This is important because it would suggest that vividness is either driven by the stimuli (if participants strongly agree on vividness ratings) or by personal evaluations (if there is little agreement).

It was expected that some images would be easier to remember/visualize than others. Item (image) was treated as a random effect in all relevant analyses, so any image-specific effect, including those related to vividness, were explicitly modeled.

That said, there was some consistency across subjects in vividness ratings applied to particular images. For each subject, the subject's vividness values for all 90 recalled images were correlated with the average vividness values across all other subjects for those images (only using those subjects that saw the image during encoding, as opposed to its pair). The mean correlation value across subjects was significantly greater than zero [$r = .18$, $p < .005$, one-tailed 200 sample permutation test]. Added result to the page 8, line 150.

2. The brain-wide decoding accuracy values should be analyzed in greater detail. In particular, in which areas do the accuracies differ between the different levels? Is there indeed higher accuracy for low-level features in low-level visual areas and for high-level features in higher-level visual areas – and how are low- and high-level visual areas defined? Is there differential accuracy in prefrontal ROIs? Shouldn't this analysis also allow to account for the correlation between the different feature levels?

The brain-wide decoding accuracy values should be analyzed in greater detail.

The revised manuscript offers a clearer explanation of our rationale for preferring FSIC to a mean-decoding approach and thus why we do not focus on mean decoding accuracy. In brief, the goal of the study was to identify feature-specific visual reactivation throughout the cortex during recall and relate this reactivation to behavior. FSIC was used instead of a mean-accuracy based approach because it is able to control for correlations between feature-levels (see page 12, lines 242-252) as well as identify reactivation during recall that the mean-accuracy approach would be blind to (see pages 12-13, lines 253-266).

Is there indeed higher accuracy for low-level features in low-level visual areas and for high-level features in higher-level visual areas – and how are low- and high-level visual areas defined?

Figure 4a indicates that the cortical accuracy peaks for each feature level are consistent with the cortical visual hierarchy. This cortical hierarchical organization was also evident in the correlations between neural reactivation, imagery vividness ratings and recognition response accuracy (Figure 5). See page 20, lines 425-435.

Is there differential accuracy in prefrontal ROIs?

Our FSIC results clearly indicate that higher-level features are more strongly represented in the prefrontal region, as expected.

3. Related to the previous point: In addition to these results, the actual accuracy values should be reported. This is particularly important since the authors trained their model on a total of only 21 minutes of visual stimulation rather than several hours of stimulus presentation in previous encoding studies.

Added Supplementary Figure 1, which contains decoding accuracy (rank measure) over the entire retrieval period averaged over all ROIs. Reference added to page 10, line 201.

4. Relationship to behavior: first of all, did the authors correct for multiple comparisons across feature levels and ROIs? This would be necessary.

We corrected for multiple comparisons across all coefficients, i.e. all four combinations of feature level and ROIs.

“Figure 5b depicts the partial regression coefficients associated with the four reactivation measures (corrected for multiple comparisons over the four coefficients using FDR).” Added to page 16, line 337.

5. The explanation in terms of predictive coding seems a bit far-fetched and very indirect. Does that mean that participants imagine two different images, one bottom-up and the other one top-down? Is there any evidence for this?

We think that our results demand an interpretation grounded in theories linking bottom-up and top-down contributions to memory. There is a large and well supported literature on predictive coding (Rao & Ballard, 1999; Friston, 2005, 2010; Bastos et al., 2012; Axmacher et al., 2010;

Henson & Gagnepain, 2010), including the top-down inference of expected visual stimuli (Muckli et al., 2015) and concurrent bottom-up and top-down representations within individual regions along the visual hierarchy (Rademaker, Chunharas, & Serences, 2019). Added the last two references to the discussion (page 24, line 510).

According to predictive coding theory, during encoding there should be at least two representations of low-level features: a bottom up representation that accurately represents the low-level features present in the presented image, and a top-down representation of low-level features that is inferred from higher-level features present in the image (the latter is compared with the former to determine if the high-level features are correct). Consequently, the top-down inferred features should be yoked to the associated high-level features, thereby enabling the decoding of high-level features within the early visual cortex when trained on encoding data.

“From this perspective, the presence of higher-level features within the lower-level ROI may represent the top-down inference of lower-level features **because these inferred features would be yoked to the associated high-level features.**” Added to results (page 17, line 351).

We argue that there are multiple ways to recall a detailed visual memory. One is to infer lower-level features from the encoded higher-level features. The benefit of this approach is that this can be done without storing the relatively uncompressed lower-level features, but at the cost of a more generic (less image-specific) memory. Conversely, the lower-level features could be encoded resulting in a more detailed and image-specific memory, but at the cost of greatly increased storage demands. We suggest in the discussion that a compromise between these two extremes is likely, with only a very sparse subset of the most diagnostic lower-level features being encoded that were chosen at the time of encoding based upon how poorly they were predicted/inferred by the higher-level features (it is unnecessary to directly encode lower-level features that are already encoded in the more compressed higher-level features). These encoded low-level features serve to constrain the inferred low-level features to a smaller subspace that better represents the visual details of the encoded image.

“We propose that during episodic memory recall, cued/recalled higher-level features are used to infer lower-level features, while the sparsely recalled lower level features that were not accurately predicted during perception serve to constrain this inference to be specific to the recalled episode (**individually storing lower-level features that are effectively stored in the more compressed higher-level features would be inefficient.**)” Added to discussion (page 24, line 519).

Reviewer #2 (Remarks to the Author):

Bone and colleagues report a human fMRI study that uses convolutional neural networks to measure representations of visual scenes at multiple feature levels and to then test for

reactivation of these scenes, at each feature level, across multiple cortical areas. The paradigm is relatively straightforward with the key component being the retrieval phase, where subjects first recall a scene and then indicate whether a probe scene is the target vs. a lure. The paper is generally well written and addresses topics of potentially broad interest.

The first major claim is that feature information, at every level, is reactivated throughout much of cortex. This is consistent with many prior studies (including from these authors) showing that reactivation is widely distributed across many cortical areas (including frontoparietal areas). There is also an analysis that is described called feature-specific informational connectivity (FSIC). This analysis is framed as a novel method, but the idea of correlating information-based measures across brain regions is not new (see work from Coutanche, as the authors note). But even the specific application of this method to measures of memory reactivation is not new (Kuhl, Johnson, Chun 2013). It is just the specific application of this method to reactivation indexed by a CNN that has not previously been reported. So, it is a fairly incremental advance. Second, reactivation at multiple levels was associated with vividness, which again is consistent with a number of prior studies

(several from these authors) that have shown that reactivation in multiple brain regions (visual cortex and beyond) is related to vividness. Third, subjects that showed stronger reactivation showed better recognition accuracy in the old/new test. The relationship between level-specific measures of reactivation and performance on the recognition memory test is potentially the most interesting aspect of the present paper, but the analyses/results are not as selective (or convincing) as they would need to be for this to support strong conclusions. Namely, the relationship between reactivation and recognition accuracy was only observed in an across subject correlation (which is a less-preferred measure than within-subject relationships) and was not selective to the specific recognition memory trials (new trials) that tested discrimination between similar memories. Collectively, the paper touches on several interesting ideas and uses very sophisticated methods. In particular, the use

of CNNs to index memory reactivation at different levels is very interesting and is likely to appeal to readers. That said, the current results ultimately represent mostly incremental advances given that there are several other studies that already report similar findings.

Major Comments

1. Figure 4b is referenced as the critical evidence that the FSIC analysis revealed feature-selective reactivation in feature-relevant cortical areas. However, there does not appear to be any direct statistical test to support this claim of selectivity. Rather, it is based on a qualitative assessment of the regions that 'showed up' in each analysis. The authors refer to the comparison of the on diagonal vs. off diagonal effects, but it does not appear that there is a direct statistical test of whether the on-diagonal effects are significantly stronger than the off diagonal effects.

This analysis may well turn out a positive effect (significant difference), but I think it is important that these claims about selectivity be based on a statistical comparison.

Thank you for pointing out this oversight. We have now added on- vs. off-diagonal comparison to the results:

“We found that the partial correlation coefficients within the diagonal were significantly greater than the coefficients within the off-diagonal [on-diagonal: mean = .075; off-diagonal: mean = .013; difference: mean = .063, 90% CI lower bound = .062, $p < .001$, one-tailed, paired samples].” (page 15, line 304).

2. Related to the above: the graph for the FSIC simulation conveys a lot more information than does the figure which shows the actual (non-simulated) results. The ‘real data’ figure just shows ROIs on brains and it is harder to assess the effect sizes, the comparison across ROIs/feature levels, etc.

Thanks for this suggestion. We have now added a bar graph of the FSIC results averaged over all ROIs to Figure 4. Note: this graph is not the same as Figure 3b—we could not replicate that figure because we do not know a priori what features are represented in the ROIs.

3. The feature specific informational connectivity analysis (FSIC) is fairly confusing. It was not at all intuitive to me how/why this measure is preferable to other approaches. And I also had trouble following exactly how it was derived (ultimately, I understood it, but it took a few reas).

We now describe the rationale for the approach more clearly, starting on page 11, line 231.

“The key insight underlying FSIC is that trial-by-trial memory fidelity will naturally vary between feature levels as a result of differences in the proportion of recalled features and the overall diagnosticity of those features (i.e. the extent to which the features can be used to separate the target image from the other recalled images). Consequently, each feature layer will be associated with a unique (but not independent) pattern of reactivation over trials. Moreover, assuming feature-specific information is shared/coordinated across regions, as suggested by the theorized role of the frontal cortex in the coordination of reactivation during recall (Mechelli, Price, Friston & Ishai, 2004; Nobre et al., 2004; Higo et al., 2011; Lee & D’Esposito, 2012; Dentico et al., 2014; Dijkstra, Zeidman, Ondobaka, Gerven, & Friston, 2017), regions that represent the same, or very similar, feature-specific information should display similar trial-by-trial reactivation patterns.

FSIC works by extracting the trial-by-trial reactivation pattern for a given feature-level from a representative seed region and subsequently looking for a significant match in a target ROI elsewhere in the cortex. Due to the correlation between features from different levels of the neural network (VGG16), as well as the expected trial-by-trial correlation in the number of

features recalled across feature levels (e.g. it is reasonable to expect that the detailed recall of low-level features is often accompanied by the detailed recall of high-level features), it was necessary to control for the trial-by-trial variability associated with non-target feature levels. To this end, FSIC measures the correlation between trial-by-trial feature-specific neural reactivation in a seed ROI and a target ROI, while regressing out the reactivation associated with all non-target feature-levels in the target ROI, thereby enabling the detection neural reactivation associated with a specific set of features.

By capturing this interregional trial-by-trial variance in reactivation fidelity, FSIC not only has the potential to be more sensitive than simply assessing mean decoding accuracy, it can also uncover reactivation that the mean-based method would not be able to capture. For example, suppose a study participant is recalling one of two images per trial. 50% of the time the subject accurately recalls the target image, while 50% of the time the subject mistakenly recalls the other (non-target) image. Let's further assume the image decoding during recall is 100% accurate. In this case, the mean decoding accuracy (for the target image) would be at chance (50%) leading to the false conclusion that the features being decoded are not present in the ROI during recall. In contrast, if we correlated trial-by-trial decoding accuracy between two regions (each with 100% decoding accuracy) we would get a perfect correlation, correctly indicating that the relevant features are represented in the ROIs. More realistically, any variance in recall fidelity (i.e. deviation from perfect recall) would reduce the power of studies using mean decoding accuracy, while not adversely affecting the power of studies using the trial-by-trial correlation method, i.e. FSIC.”

4. More specifically, the Results section that that the FSIC analysis covaries out all non-target feature levels. Elsewhere it describes the analysis as “controlling for all non-target feature levels”. But it is not clear to me how (or why) this analysis controls for the correlation across feature levels. Was there some step taken in this analysis other than just correlating the ranks? The methods section does not provide any clarifying detail about this.

“For the FSIC analyses, partial regression coefficients were calculated (using trial-by-trial reactivation data from the recall period) with separate linear mixed-effects (LME) models for all combinations of subject, seed ROI, and target ROI. For each LME model, reactivation (rank measure) of the associated feature level for the seed ROI was the dependent variable (DV), reactivation for each of the four feature levels within the target ROI were the independent variables (IV), and participant and image were crossed random effects (random-intercept only, due to model complexity limitations).” Updated page 40, line 858.

5. Also, my understanding is that FSIC approach doesn't actually test whether reactivation is “above chance” but only tracks whether two regions represent similar information (whether or not that information is accurate). Assuming this interpretation is correct, this should be made

clearer and claims should be adjusted accordingly. For example, if FSIC is only tracking similarity of information across regions then, on its own, it does not establish whether or not feature level reactivation is present. Rather, two regions could each have completely inaccurate feature-level reactivation, but these 'representations' could be highly correlated across regions.

With regard to the rationale for the FSIC method, see our response to comment 3.

We appreciate the reviewer raising the possibility that:

"two regions could each have completely inaccurate feature-level reactivation, but these 'representations' could be highly correlated across regions." We do not believe this phenomenon is driving our results for the following reason. Let's suppose that activity independent of visual features is correlated across regions (i.e. noise correlations). Could this be the cause of, for example, the low-level representations FSIC detected in the frontal cortex? If this was the case, then we should see very similar results across feature-levels if the same seed ROI is used. Despite using the same ROIs as the low-level seed, the results for mid-level features within the frontal cortex depicted in Supplementary Figure 3c differ greatly from the low-level results depicted in the top-left of Figure 4b, providing strong evidence that our findings are not the result of noise correlations.

The above reasoning was added to pages 26-27 lines 565-573.

6. The data in 5b is from a within-subject analysis whereas the data in 5c is from a between subjects analysis. This switching between different kinds of measures is not ideal. And the selectivity of the relationship in 5b to the low-level features is undermined a bit by the fact that the relationship for the low-level features is likely not significantly stronger than the relationship for the high-level features (though, this statistic is not reported).

With respect to: *The data in 5b is from a within-subject analysis whereas the data in 5c is from a between subjects analysis. This switching between different kinds of measures is not ideal.*

We have now added the within-subject result for the high-accuracy group to Figure 5(d).

And the selectivity of the relationship in 5b to the low-level features is undermined a bit by the fact that the relationship for the low-level features is likely not significantly stronger than the relationship for the high-level features (though, this statistic is not reported).

We addressed the—unexpected—nearly identical regression coefficients for lower- and higher-level features in the discussion (Feature-Specific Neural Reactivation and Memory Vividness).

We have included statistics for the difference between coefficients for lower- and higher-level features for vividness:

“However, in contradiction to our second prediction, the lower-level partial correlation coefficient was not significantly greater than the higher-level coefficient [lower-level coefficient - higher-level coefficient: .005, $p = .423$, one-tailed, paired samples], indicating that both lower and higher-level features contribute approximately equally to subjective vividness.” (page 16, line 341)

between-subject accuracy:

“...this correlation was significantly greater than the correlation with higher-level features in the higher-level ROI [lower-level coefficient - higher-level coefficient: 1.199, $p = .032$; one-tailed, paired samples].” (page 18, line 377)

And within-subject accuracy:

“...but it was not significantly greater than the higher-level coefficient [lower-level coefficient - higher-level coefficient: .050, $p = .155$; one-tailed, paired samples].” (page 19, line 402)

7. For the analyses relating reactivation to recognition memory (which, again, I think is an interesting analysis), the primary/initial analyses are based on overall recognition memory (old/new) accuracy. However, it would seem that the predictions should be very different for the old vs. the new trials. The authors are testing the idea that only low-level reactivation will predict recognition accuracy with the idea that low level details are necessary to discriminate the new items (which are very similar to old items) from the old items. But subjects are not actually selecting between old and new items—rather, they are making old/new judgments EITHER about old or new items. For the old trials, it seems that reactivation at any level (high OR low) would be likely to lead subjects to (correctly) endorse the item as ‘old’ (hits). But, for the new trials, high-level reactivation of semantic information would potentially lead subjects to erroneously endorse the new items as old (false alarms) whereas low-level details should be helpful in rejecting the new item (correct rejection). Thus, it would seem that to test their critical prediction, the authors should restrict all of these analyses to the new trials. This is done for a couple of the post-hoc analyses, but it seems like this should be the key focus of all of the recognition memory analyses.

The reviewer raises a good point. We originally included both old and new trials because we had no a priori assumption that the relationship between accuracy and reactivation would be different between old and new trials. Our thinking was that since the participants knew that the lure would be semantically very similar to the memorized image (due to their experience in the practice trials and previous recall trials), they should rely disproportionately upon low-level features when making their choice (whether the image was old or a lure), because the similarity of high-level/semantic features is not usually diagnostic. The tendency for a given participant to rely on higher-level features can be measured by looking at the participants' average "lure" trial accuracy, because a bias towards using higher-level features should result in lower "lure" trial accuracy (note: we did not find significantly lower accuracy on lure trials; added to page 8, line 160). To

address the reviewer's point, we related participants' average "lure" trial accuracy to their trial-by-trial correlation between accuracy and reactivation. As noted in the paper, our results indicated that only some subjects (those with higher average "lure" accuracy) showed the expected correlation, which we interpreted as evidence that only some subjects relied upon low-level features (to the other subjects' detriment). See Figure 5 and Supplementary Figure 6.

8. Also, even if the issues above are ignored, it is not clear that the relationship between low-level reactivation and recognition accuracy is STRONGER than the relationship between high-level reactivation and subsequent memory. In other words, there are some effects for low but not high vividness trials, but in order to support the prediction that low level reactivation is MORE IMPORTANT, this would require a statistical difference between the coefficients for low vs. high level reactivation.

This is an excellent point. We have now included statistics for the difference between coefficients for lower- and higher-level features for both vividness and accuracy (page 16, line 341; page 18, line 377; page 19, line 402).

For the vividness results, we addressed the nearly identical regression coefficients for lower- and higher-level features in the discussion (Feature-Specific Neural Reactivation and Memory Vividness).

For the accuracy results, the difference was significant for the between-subject measure, but not the within-subject measure.

9. The authors repeatedly emphasize the importance of feature specific representations and “decomposing” memory reactivation. There are two recent papers on these specific topics that should be discussed (or at least cited): Favila, Samide, Sweigart, & Kuhl, 2018; Lee, Samide, Richter, & Kuhl, 2018. The paper by Favila specifically considers feature-level memory reactivation across cortical areas and the relation of these representations to behavior. And the paper by Lee explicitly discusses “decomposing memory reactivation” in order to test how/whether different levels of reactivation predict accuracy in a memory test that includes targets and highly similar lures (exactly as in the current study). However, in the paper by Lee et al., analyses were separated for old and new trials (see Comment 7) and it was found that for the new items (which are the critical trials) more generic reactivation predicted false alarms whereas more detailed reactivation predicted correct rejections. The similarity of these prior papers to the current work also reduces the novelty / conceptual advance of the current study.

Thanks for pointing out this oversight and we now cite these papers and briefly point out differences in our approach.

Favila, Samide, Sweigart, & Kuhl, 2018:

The authors investigated reactivation of color and object category in the occipitotemporal cortex and the lateral parietal cortex. The primary goal for the current study was to assess the cortical distribution of features from multiple levels of the visual hierarchy (controlling for the non-target feature-levels) throughout the entire neocortex. We were particularly interested in whether representations of low-level visual features (e.g. edges) exist within the frontal cortex and are reinstated during recall—a question that is not addressed by Favila, Samide, Sweigart, & Kuhl, 2018.

Reference added to page 4, line 78.

Lee, Samide, Richter, & Kuhl, 2018:

The authors here certainly address a related question. However, in Lee et al. the categories used in the study were not feature-specific (a key component of our study). The only features explicitly represented were categorical features (“face”, “house”, “object”), which was contrasted with item/image-specific representations. As the authors note:

"While it is tempting to interpret this dissociation as evidence that category reactivation maps to semantic reactivation whereas item reactivation maps to perceptual reactivation, this is likely an oversimplification. Indeed, it is more likely that each measure includes a combination of semantic and perceptual information, though perhaps to varying degrees."

Additionally, the early visual cortex was not included in the analysis (only the angular gyrus, medial parietal cortex and ventral temporal cortex were examined). Consequently, while the study is a valuable contribution to our understanding of the neural basis of memory specificity, it does not specifically address the behavioral contribution of low-level visual features to recognition memory—the focus of the recognition section of our paper.

“(although previous work (Lee, Samide, Richter, & Kuhl, 2018) has shown that recognition accuracy is predicted by item/image-specific neural reactivation, there is no direct evidence that the finding was due to the reactivation of low-level features).” Reference added to page 25, line 535.

Minor Comments

10. Related to the analysis in 5c, I was confused by the statement that “within- and between-subject coefficients were pooled together.” I thought this analysis only used the averaged within-subject effects—in other words, how could there be within-subject coefficients if the within-subject data was averaged together?

The coefficients were not averaged together, rather, when computing statistical significance via FDR the p-values associated with each coefficient were pooled together so as to correct for multiple comparisons over all tests. Updated page 18, line 379.

11. The seed ROI selection section (line 744) is very confusing. In part it is hard to follow the steps that were actually taken, but it is even less clear why these steps are being taken.

We have tried to clarify the procedure. “The goal of the ROI seed selection was to identify the ROIs with the greatest reactivation for target feature level relative to the non-target feature levels, controlling for mean reactivation across ROIs. The procedure for generating weight values for each ROI (Figure 3a) was as follows: 1) get the average classification accuracy across subjects during image perception (data taken from the old/new recognition task during the retrieval blocks) for each feature-level and ROI. 2) z-score classification accuracy across ROIs for each feature level, thereby controlling for differences in mean reactivation (across ROIs) between feature levels. 3) set all values less than zero to zero, so that ROIs with z-scores less than zero (i.e. ROIs with relatively low reactivation of the target feature-level) would not be assigned a non-zero weight. 4) for each ROI and feature-level, subtract the greatest value associated with the other feature levels from the target feature’s value. 5) set all values less than zero to zero. As a result of steps 4 and 5, only those ROIs that show greater relative reactivation for the target feature level than all other feature levels will have a non-zero weight, and this weight will be proportional to the difference between the relative reactivation of the target feature level and the greatest non-target feature level. 6) normalize the values across ROIs to sum up to one (i.e. divide each value by the sum of all values) for each feature level. 7) set values less than .05 to 0 to retain only those ROIs that were assigned a non-negligible weight (This was done so that more non-seed ROIs could be included in the FSIC analysis). 8) normalize the values across ROIs for each feature level again, because the weights will no longer sum to one if any weights were set to zero in step 7.”

Updated “Seed ROI Selection”.

12. While I think there is reason to have some faith that the layers taken from the CNN to represent early, mid, late visual and semantic information are meaningful, it should be noted that these are not independently validated (e.g., through behavioral analyses). Put another: how do we know that these particular layers map onto psychologically-relevant early, mid, late, and semantic levels? I don’t view this as a major problem, but it might be something that could be briefly touched on in the Discussion (i.e., what assumptions are being made in interpreting the CNN layers?).

There is strong evidence that the layers of DNNs map onto psychologically-relevant early, mid, late, and semantic levels. Güçlü and van Gerven (2015) showed that the low, mid and high-level layers of a DNN preferentially represent low-level (blob, contrast, and edge), mid-level (contour, shape, and texture), and high-level (irregular pattern, object part, and entire object) feature classes, respectively (Figure 3 from their paper). Moreover, they showed that the hierarchical structure of the DNN layers are reflected in their cortical representation, i.e. features from low-level DNN layers were most strongly represented in the early visual cortex, while features from high-level DNN layers were most strongly represented in the lateral occipital cortex (Figure 4 from their paper).

13. Line 725: “For each value of the regularization parameter, the model parameters h were estimated for each voxel and then prediction accuracy (sum of squared errors; SSE) of the recognition data was measured using 3-fold cross validation. For each voxel, the model parameters h that yielded the highest prediction accuracy were retained for image decoding.” I found this a bit confusing. I assume the recognition data is the old/new portion of the retrieval blocks? How exactly was the 3-fold cross validation implemented? And why was the recognition data used for this part (as opposed to the encoding trials)?

Thanks for pointing this out. This description was incorrect and described an earlier version of the analysis. For the version in the current study, we used the held-out encoding data when selecting the regularization parameter. The correct description has been added to the Encoding Model section. As an aside, the earlier approach excluded the recognition trials that contained the to-be-predicted image or its pair by extending the 3-fold cross-validation (which was performed over images, not trials: clarification added to page 38, line 812) to the retrieval trial selection.

14. Perhaps I missed this somewhere, but was the significance of the reactivation measures assessed by permutation tests?

Bootstrapping was used (see Methods: Feature-Specific Informational Connectivity (page 40, line 864), and Linear Models and Statistics).

Reviewer #3 (Remarks to the Author):

Bone and colleagues investigate how activity predicted from various hierarchical layers of a deep convolutional neural network (CNN) maps onto the brain activity patterns recorded with fMRI during a memory recall task, where participants are asked to vividly imagine previously seen images. They develop a method called feature-specific informational connectivity (FSIC) that effectively regresses out features that are correlated between layers, as demonstrated with simulated data. Apart from this methodological aspect, the paper also describes a number of interesting empirical findings. Most interestingly maybe, low-level visual features of the images are not confined to low-level visual cortex during recall/imagery, but are represented throughout high-level areas including parietal and frontal lobes. They also find that the activation of both high and low-level features contributes roughly equally to the vividness of mental imagery. Somewhat less robustly, low-level feature

reactivation is shown to be beneficial for discriminating a previously seen target image from a highly similar lure image.

This was a very interesting read, well written despite the complexity of the methods. The paper could make a novel contribution to the literature both from a methodological and an empirical perspective. I have a number of comments, however, that should be addressed in a revision.

Methodological comments:

(1) The pure recall/imagery part of each retrieval trial was followed relatively closely in time (7 sec) by visual presentation of a recognition stimulus (an old target or a very similar lure). The authors state in the methods section that they separately model the hemodynamic responses to recall and recognition onsets. However, modelling both contributions does not rule out that the recall regressor picks up variance shared with the recognition regressor, and there is thus a concern that the main results are partly produced by an overlap in the sensory input between training and test data, and not by pure recall/mental imagery activity. This concern could be addressed by regressing out the shared variance from the cued recall beta coefficients.

The reviewer is correct about this and it was also a concern of ours. We did indeed account for the shared variance on a trial-by-trial basis:

"Estimates of beta coefficients for each trial and task were computed via a separate linear regression per trial (each with 16 samples: one per time point), with vertex activity as the dependent variable, and the expected hemodynamic response values for the "recall" and "old/new judgment" tasks as independent variables." (page 34, line 729)

There was one linear model for a given trial with two independent variables: one for "recall" and another for "old/new judgment".

(2) Related to the first point, the authors state in the last sentence of the “Encoding models” section (l. 725-29) that the encoding models were optimized for predicting the recognition data. This statement implies that model training already involved the recognition phase data, which are naturally highly correlated with the cued recall (test) data. This could cause a major confound and should be clarified.

Thanks for raising this important issue. The description included in the previous version of our manuscript was incorrect and described an earlier version of the analysis. For the version in the current study, we used the held-out encoding data when selecting the regularization parameter. The correct description has been added to the Encoding Model section (page 38, line 808). As an aside, the earlier approach excluded the recognition trials that contained the to-be-predicted image or its pair by extending the 3-fold cross-validation (which was performed over images, not trials: clarification added to page 38, line 812) to the retrieval trial selection.

(3) In the selection procedure for seed ROIs, what is the reason for setting below zero values to zero in steps 3 and 5, and why was it necessary to normalize the values twice (steps 6 and 8)? These steps should be justified in the methods section.

Clarified the Seed ROI section (page 44).

...what is the reason for setting below zero values to zero in steps 3 and 5...

3) set all values less than zero to zero, so that ROIs with z-scores less than zero (i.e. ROIs with relatively low reactivation of the target feature-level) would not be assigned a non-zero weight.

5) set all values less than zero to zero. As a result of steps 4 and 5, only those ROIs that show greater relative reactivation for the target feature level than all other feature levels will have a non-zero weight, and this weight will be proportional to the difference between the relative reactivation of the target feature level and the greatest non-target feature level.

...why was it necessary to normalize the values twice (steps 6 and 8)?

6) normalize the values across ROIs to sum up to one (i.e. divide each value by the sum of all values) for each feature level.

7) set values less than .05 to 0 to retain only those ROIs that were assigned a non-negligible weight (This was done so that more non-seed ROIs could be included in the FSIC analysis).

8) normalize the values across ROIs for each feature level again, because the weights will no longer sum to one if any weights were set to zero in step 7.

(4) The rationale for using the videos for training is not entirely clear, and deserves a few sentences in the methods section. In particular, did these videos contain all the 90 stimulus concepts used in the later encoding/retrieval parts?

The videos were included in the study for a different experiment focused on hyperalignment (Haxby et al. 2011). We used the videos in this study to provide additional training data.

“The training clips/images did not contain all the object categories of the 90 images used in the encoding/retrieval parts... Consequently, some relevant semantic features may not be effectively mapped onto brain activity. To address these issues, VGG16's softmax output layer (the last layer of the CNN) was chosen to represent visual object semantics because it... [represents] categorical confusion. Because related object categories are often confused with each other in deep CNNs (e.g. "grille" confused with "convertible"; for more examples see figure 4 in Krizhevsky, Sutskever, & Hinton, 2012), the inclusion of these categorical errors... [captures] broader (less exact) object semantics. This enables semantic feature-brain mappings to be learnt by the encoding models when the training set contains images that are semantically related to the test set images... However, networks can also confuse visually similar, but semantically unrelated categories, increasing the likelihood that semantic and (high-level) visual features will be conflated. This potential confound is addressed by controlling for the correlations between feature levels—a focus of the current study.” Added to page 41, page 827-844.

(5) Minor clarification comment: the authors should state whether they used Spearman or Pearson R for correlating predicted activity with recall activity.

Pearson. Added to Methods: Image Decoding (page 39, line 829)

Results:

(6) When reporting the behavioral data, the authors should report separately the hit rates and misses to old target items, and the correct rejections and false alarms to new similar lure images. It would also be interesting to see the relationship of feature reactivation of the different layers specifically on trials where participants made false alarms, contrasting the maps with correct old judgments. The idea being that a very generic (high-level) reactivation will be related to a higher likelihood of judging a similar lure as old (as found e.g. in Lee et al., 2018, CerebCortex). I understand though if not enough trials are available to separately investigate these conditions.

Thanks for alerting us to this oversight. We now provide this information: “Accuracy on “old” and “new”/lure trials was 79.2% (SD = 12.3%) and 82.8% (SD = 13.4%), respectively, with no significant difference in accuracy between the two conditions ($t(26) = 1.34$, $p = .193$, paired samples, two-tailed).” Added to page 8, line 160.

We initially performed the analyses the way we did because we had no a priori assumption that the relationship between accuracy and reactivation would be differ between old and new trials. We assumed that participants knew that the lure would be semantically very similar to the memorized image due to their experience in the practice trials and previous recall trials. Consequently, we expected all participants to rely disproportionately upon low-level features when making their choice (whether the image was old or a lure), because they knew that similarity of high-level/semantic features is not usually diagnostic. The tendency for a given participant to rely on higher-level features can be measured by looking at the participants' average "lure" trial accuracy, because a bias towards using higher-level features should result in lower "lure" trial accuracy (note: we did not find significantly lower accuracy on lure trials). To address the reviewer's question, we related participants' average "lure" trial accuracy to their trial-by-trial correlation between accuracy and reactivation. As noted in the paper, our results indicated that only some subjects (those with higher average "lure" accuracy) showed the expected correlation, which we interpreted as evidence that only some subjects relied upon low-level features (to the other subjects' detriment). See Figure 5 and Supplementary Figure 6.

(7) It would be very helpful if a supplementary figure could be included showing the model fits for the different layers during the video/encoding runs (i.e., using the cross-validation results). Such a figure would allow the reader to directly compare layer-specific predictions when the model is tested on visual perception versus memory recall. This is particularly interesting with respect to the surprising finding that the mid-layers of the CNN map onto occipital lobe only during recall.

We didn't include encoding model fits because FSIC might not function as intended when analyzing perception data (and using the mean reactivation results would be misleading due to the correlation between feature levels). FSIC relies on trial-by-trial variation in reactivation that is shared across ROIs, and the expected source of this variation during recall is variation in memory fidelity. Specifically, variation associated with the number of features forgotten, and variation associated with which features are forgotten (some features are more diagnostic than others; clarification added to the paragraph on page 11, line 231). These are the two sources of variance that were modeled in Fig 3b (as an aside, our simulation results indicated that only the latter source, which features are forgotten, was necessary to control for the non-target feature levels because the former source, the proportion/number of features forgotten, was set to be identical across feature-levels). Attention during perception/encoding may provide similar sources of variance (e.g. the number of features perceived, and which features are perceived), but this is unclear. Future work, which is beyond to scope of the current paper, will be necessary to explore whether, and in what experimental contexts, FSIC can be applied outside of recall.

(8) Do the authors have information about what layers in their network are most sensitive to spatial frequency of the images? Fig. 1c suggests that spatial frequency differs between layers.

Apart from the perceptual-to-semantic dimension which is the focus of the discussion in the paper, an additional detail-to-gist dimension might explain some of the differences in the main results, and is worth discussing.

The spatial resolution was equalized across layers by subsampling all convolutional layers (the low-, mid- and high-level visual layers) to 3x3. The images in figure 1c were not subsampled to display the differences in feature representations. Also, because higher-level features are composed of lower-level features it is more accurate to state that higher-level features are more translation/location-invariant than lower-level features; they are not necessarily less sensitive to spatial frequency. It is nevertheless possible that the detail-to-gist dimension is related to the findings we observe, however, it is difficult to map this concept directly on to our findings, which are based on the output of a specific multi-layered convolutional neural network. We do think that it is an important area for future research and a future opportunity for gaining insight into the nature of higher-level neural representations.

(9) The result that mid-layer representations are constrained to occipital lobe is quite surprising. The authors should discuss whether there is something particular about the 7th layer e.g. in terms of spatial frequency (see previous comment). If they already have the results also for neighbouring layers 6 and 8, it would be interesting to hear whether the same result is true for those layers (but to be clear, this is not a requested re-analysis).

We are not sure what the cause is. It shouldn't relate to spatial frequency for the reasons stated above. It may be the case that mid-level features are redundant when both lower-level and higher-level features are present in a region, and low-level features provide more useful information not present in the higher-level features—as detailed in the discussion. We add the following to the revised manuscript:

“Conversely, the relative lack of mid-level features within these higher-order regions may indicate that such features are largely redundant/uninformative given the presence of both low- and high-level features.” Added to page 23, line 487.

To examine the robustness of our results to the selection of CNN layers, we performed a variant of our FSIC analysis (Figure 4b) using all layers. We first subdivided the sixteen layers into four groups, with convolutional layers 1-4, 5-9, 10-13 and fully connected layers 14-16 assigned to the low-, mid-, high-visual and semantic groups, respectively. The reactivation values for each of these layers were then averaged, producing a single reactivation value for each of the four groups for each combination of subject, ROI and trial. FSIC was then performed using these values in a manner identical to the original approach. The results (Supplementary Figure 3) were very similar to the original results (Figure 4b), including mid-level features, suggesting the results are robust to layer selection.

Conceptual:

(10) I find it somewhat misleading to call the method developed here a connectivity method, because (i) similarity/correlation between two areas does not imply connectivity, and (ii) the method effectively regresses out the overlap with other ROIs, and thus seems to amplify dissimilar patterns. This should be clarified.

Because the method we use is a derivation of “informational connectivity” (Countanche et al. 2013) we decided to preserve the original terminology. Moreover, the method is a connectivity method the way “functional connectivity” is a connectivity method, in the sense that it measures coupling between two time-series (here, an informational time-series) between multiple regions. Because we use multiple covariates, we are effectively partialing out other sources of variance. But this is also commonly done in “functional connectivity” analyses (e.g. psychophysiological interactions, or in the case of functional connectivity with nuisance regressors, etc.). FSIC does not strictly speaking regress out the overlap with other ROIs; rather, it regresses out the reactivation of non-target feature-levels within a given ROI. See the paragraph on page 12, line 242.

(11) A number of papers should be discussed in the context of the present study, including (i) Dijkstra et al. (2018, eLIFE) suggesting it is mainly the late visual processing components that are active during later imagery; (ii) Linde-Domingo et al. (2019, Nat Comm) suggesting that higher-level semantic information is reactivated more rapidly during recall than lower-level perceptual information; (iii) Martin et al. (2018, eLIFE) suggesting that integrated low- and high-level features are represented at very late stages of the visual processing hierarchy. In addition, it might be worth discussing how FSIC relates to other recently developed information-theoretical frameworks as published in Zhan et al. (2019, Curr Biol).

Thank you for these suggestions and we are aware of these excellent papers, some of which were published very recently.

Dijkstra et al. (2018, eLIFE):

The findings of Dijkstra et al. (2018, eLIFE) are somewhat relevant to our study. Our study was concerned with feature-specific reactivation, whereas Dijkstra et al. examined the temporal dynamics of reactivation. Dijkstra et al. decoded stimulus category (face, house). Consequently, the study was blind to the features being reactivated. Higher-level features must not be conflated with “[temporally] late visual processing components”; as the authors state:

"There was no overlap between imagery and perceptual processing until 130 ms after stimulus onset, when the feedforward sweep is presumably completed and high-level categorical information is activated for the first time. One explanation for this discrepancy is that representations in low-level visual areas first have to be sharpened by feedback connections (Kok et al., 2012) before they have a format that is accessible by top-down imagery."

i.e., the late visual processing components could have included (sharpened) representations of low-level features. This interpretation is consistent with the predictive coding account of visual perception and memory recall covered in the discussion.

Reference added to page 4, line 76.

Linde-Domingo et al. (2019, Nat Comm):

This paper also concerns the temporal properties of reactivation. The authors found higher-level semantic information (animate vs. inanimate) is reactivated more rapidly during recall than lower-level perceptual information (photo vs. line drawing), whereas the opposite was found during perception.

Reference added to page 4, line 78.

Martin et al. (2018, eLIFE):

Martin et al. scanned subjects while they judged whether a visual (e.g. 'round', 'light') or conceptual (e.g. 'animate', 'tool') word accurately described an object-concept (e.g. 'gun', 'comb'), i.e. subjects recalled general properties of generic object-concepts. This is in contrast with our study, in which subjects recalled previously seen images in as much detail as possible. Answering the visual (and conceptual) judgement questions would have only required the creation of a partial low-detail mental image, or potentially no mental image at all if the descriptive-concept has a sufficiently strong semantic link to the object-concept, thereby reducing the need to engage the visual cortex relative to the detailed recall task performed in the current study (although the visual cortex did represent visual features: figure 9a in in Martin et al., 2018). Moreover, the visual representations were defined by a representational dissimilarity matrix generated from subjective visual similarity judgements (e.g. 'gun' and 'hairdryer' are similar because people judge that they have a similar shape). As a result, the representation of low-, mid- and high-level visual features were conflated, and visual features that did not directly contribute to subjective visual similarity ratings would have been overlooked. Consequently, the study's finding of overlapping visual and conceptual representations within the perirhinal cortex have limited relevance to our finding of low-level visual features within multiple higher-order regions—including the frontal cortex—during visual recall.

Reference added to page 22, line 461: "... (although, not completely unprecedented: Martin et al. (2018) identified one higher-order region, the perirhinal cortex, that contained both visual and conceptual representations while subjects judged whether visual properties (e.g. 'smooth')

matched object-concepts (e.g. ‘gun’), but the visual representations were not necessarily low-level).”

Zhan et al. (2019, *Curr Biol*):

The technique used in Zhan et al. is not directly relevant to our paper because it does not distinguish between low- and high-level visual features. Zhan et al. use an image-masking approach to link brain activity and behavioral decisions to image features, allowing the researchers to track the brain representation of task-relevant/irrelevant features over time during a perception task (features were defined by a combination of location and frequency). To distinguish between low- and high-level visual features it would be necessary to selectively remove low-level features without removing the corresponding high-level features, and it is not clear how this could be done using a simple masking approach.

References:

- Arsenault, J. S., & Buchsbaum, B. R. (2015). Distributed neural representations of phonological features during speech perception. *Journal of Neuroscience*, 35(2), 634-642.
- Arsenault, J. S., & Buchsbaum, B. R. (2016). No evidence of somatotopic place of articulation feature mapping in motor cortex during passive speech perception. *Psychonomic bulletin & review*, 23(4), 1231-1240.
- Axmacher, N., Cohen, M. X., Fell, J., Haupt, S., Dümpelmann, M., Elger, C. E., Schlaepfer, T. E., Lenartz, D., Sturm, V. & Ranganath, C. (2010). Intracranial EEG correlates of expectancy and memory formation in the human hippocampus and nucleus accumbens. *Neuron*, 65(4), 541-549.
- Bastos, A. M., Usrey, W. M., Adams, R. A., Mangun, G. R., Fries, P., & Friston, K. J. (2012). Canonical microcircuits for predictive coding. *Neuron*, 76(4), 695-711.
- Bone, M. B., St-Laurent, M., Dang, C., McQuiggan, D. A., Ryan, J. D., & Buchsbaum, B. R. (2019). Eye-movement reinstatement and neural reactivation during mental imagery. *Cerebral Cortex*, 29(3), 1075–1089.

- Coutanche, M. N., & Thompson-Schill, S. L. (2013). Informational connectivity: identifying synchronized discriminability of multi-voxel patterns across the brain. *Frontiers in human neuroscience*, 7, 15.
- Dentico, D., Cheung, B. L., Chang, J. Y., Guokas, J., Boly, M., Tononi, G., & Van Veen, B. (2014). Reversal of cortical information flow during visual imagery as compared to visual perception. *Neuroimage*, 100, 237-243.
- Dijkstra, N., Zeidman, P., Ondobaka, S., Gerven, M. A. J., & Friston, K. (2017). Distinct top-down and bottom-up brain connectivity during visual perception and imagery. *Scientific reports*, 7(1), 5677.
- Du, Y., Buchsbaum, B. R., Grady, C. L., & Alain, C. (2016). Increased activity in frontal motor cortex compensates impaired speech perception in older adults. *Nature communications*, 7, 12241.
- Friston, K. (2005). A theory of cortical responses. *Philosophical Transactions of the Royal Society of London B: Biological Sciences*, 360(1456), 815-836.
- Friston, K. (2010). The free-energy principle: a unified brain theory?. *Nature reviews neuroscience*, 11(2), 127.
- Fukushima, K. (1980). Neocognitron: A self-organizing neural network model for a mechanism of pattern recognition unaffected by shift in position. *Biological cybernetics*, 36(4), 193-202.
- Güçlü, U., & van Gerven, M. A. (2015). Deep neural networks reveal a gradient in the complexity of neural representations across the ventral stream. *Journal of Neuroscience*, 35(27), 10005-10014.
- Haxby, J. V., Guntupalli, J. S., Connolly, A. C., Halchenko, Y. O., Conroy, B. R., Gobbini, M. I., Hanke, M., & Ramadge, P. J. (2011). A common, high-dimensional model of the representational space in human ventral temporal cortex. *Neuron*, 72(2), 404-416.
- Henson, R. N., & Gagnepain, P. (2010). Predictive, interactive multiple memory systems. *Hippocampus*, 20(11), 1315-1326.

- Higo, T., Mars, R. B., Boorman, E. D., Buch, E. R., & Rushworth, M. F. (2011). Distributed and causal influence of frontal operculum in task control. *Proceedings of the National Academy of Sciences*, 108(10), 4230-4235.
- Horikawa, T., & Kamitani, Y. (2017). Generic decoding of seen and imagined objects using hierarchical visual features. *Nature communications*, 8, 15037.
- Johansson, R., Holsanova, J., Dewhurst, R., & Holmqvist, K. (2012). Eye movements during scene recollection have a functional role, but they are not reinstatements of those produced during encoding. *Journal of Experimental Psychology: Human Perception and Performance*, 38(5), 1289.
- Krizhevsky, A., Sutskever, I., & Hinton, G. E. (2012). Imagenet classification with deep convolutional neural networks. In *Advances in neural information processing systems* (pp. 1097-1105).
- Lee, T. G., & D'Esposito, M. (2012). The dynamic nature of top-down signals originating from prefrontal cortex: a combined fMRI-TMS study. *Journal of Neuroscience*, 32(44), 15458-15466.
- Mechelli, A., Price, C. J., Friston, K. J. & Ishai, A. (2004). Where bottom-up meets top-down: neuronal interactions during perception and imagery. *Cerebral Cortex*, 14, 1256-65.
- Muckli, L., De Martino, F., Vizioli, L., Petro, L. S., Smith, F. W., Ugurbil, K., Goebel, R., & Yacoub, E. (2015). Contextual feedback to superficial layers of V1. *Current Biology*, 25(20), 2690-2695.
- Nobre, A. C., Coull, J. T., Maquet, P., Frith, C. D., Vandenberghe, R., & Mesulam, M. M. (2004). Orienting attention to locations in perceptual versus mental representations. *Journal of cognitive neuroscience*, 16(3), 363-373.
- Rademaker, R. L., Chunharas, C., & Serences, J. T. (2019). Coexisting representations of sensory and mnemonic information in human visual cortex. *Nature neuroscience*, 1.
- Rao, R. P., & Ballard, D. H. (1999). Predictive coding in the visual cortex: a functional interpretation of some extra-classical receptive-field effects. *Nature neuroscience*, 2(1), 79.

Seeliger, K., Fritsche, M., Güçlü, U., Schoenmakers, S., Schoffelen, J. M., Bosch, S. E., & van Gerven, M. A. J. (2018). Convolutional neural network-based encoding and decoding of visual object recognition in space and time. *NeuroImage*, 180, 253-266.

Reviewers' comments:

Reviewer #1 (Remarks to the Author):

This is the first revision of the manuscript „Feature-Specific Neural Reactivation during Episodic Memory“. The authors present various new analyses and interpretations of their results. I think the manuscript has been substantially improved, though some crucial points remain to be addressed.

Methods

1: The authors still need to show that their approach captures retinotopic representations. Their new results only show that an analysis with a higher resolution performs worse. They would need to show that activity at individual voxels provides significant information about the visual content at specific locations on the screen, and that adjacent voxels provide information about adjacent screen locations.

2: Here it would be great if the authors could provide some results – maybe from a new behavioral study – that validates their approach: after the rating, they could ask participants to describe as many details as possible. This would allow them to interpret these ratings more objectively.

3-9: Ok

10, 11: I am still not convinced that the network indeed produces meaningful results when presented with objects that are very different from the ones it has been trained on. The authors need to demonstrate that (and to which degree) there is indeed a semantic relationship between the objects they presented (including people etc.) and the labels the network assigns, which are a subset of those it has been trained on.

12, 13: ok

Results

1,2: ok

3: The authors should test whether these accuracies are higher than would be expected by chance (i.e., higher than in label-shuffled surrogates)

4,5: ok

Reviewer #2 (Remarks to the Author):

The authors were responsive to the comments that I and the other Reviewers raised. Some of the new analyses help address concerns that were raised, though other concerns still stand. For example, the data relating reactivation measures to behavior, while interesting, remain a bit underwhelming.

The authors have also better explained the FSIC method, but I still find the approach to be non-intuitive—or, better put, to be a non-direct measure of reactivation. The authors argue that using this informational connectivity approach helps address, for example, situations where subjects might

remember the wrong image. They argue that, if subjects remember the wrong image on a given trial, traditional decoding would fail to detect any reactivation, but FSIC only cares about whether two regions have common information, whether or not that information is correct. Again, this is an indirect way to measure reactivation since the content of the reactivation is not actually verified (i.e., that it matches perception).

The obvious concern is that correlations between regions just reflect voxels within pairs of ROIs that have correlated activity (or noise correlations). The authors emphasize that if the correlations between regions were just noise correlations, then they would not differ as a function of the feature levels. This does seem to be an important point. While it still does not render the FSIC measure a “direct” measure of reactivation, I do agree that noise correlations should not differ by feature level. However, one point I am hoping the authors can clarify is exactly how non-target feature levels were regressed out. Obviously, it is critical that this regressing out of non-target feature levels isn’t what produces the selective FSIC measures for matching seed/feature levels. As an example, when considering the high-level ROI seed and the mid-level features, was the “target feature” the high-level features (i.e., relevant to the seed ROI) or was it the mid-level features? Depending on how this was done, the selectivity to the relevant feature level could simply be an artifact of regressing-out the non-target feature (but hopefully this wasn’t the case).

Reviewer #3 (Remarks to the Author):

In the revision, Bone et al. made a number of changes to the manuscript, addressing many of the concerns raised by the reviews. These changes were mainly done by re-writing or addition sections, while few new datapoints or analyses were added since the first version.

The editor asked to comment on the issue of novelty raised by the previous reviews. I do agree that there is relatively little conceptual advance from this paper, in the sense that I did not learn anything substantially new about episodic memory or how memories are reactivated in the brain. For me, the main novelty lies in the proof of principle that CNNs are able to predict neural activity patterns during memory recall periods with little bottom-up input, and that the resulting mapping onto higher- and lower-level layers of the networks shows a meaningful relationship to behaviour. I believe that many memory researchers will find this useful and feel encouraged to use such models to test more specific predictions. I thus agree with the authors’ conclusion that the results “show the potential for FSIC, and other feature-specific approaches that can decompose neural pattern representations, to test and elucidate the mechanisms [...]”.

Many of my previous comments were sufficiently addressed in the revision. Below are those comments where I was not convinced by the authors’ rebuttal.

Previous point (1), regarding the high correlation between the recall and recognition periods of a trial,

and how the authors accounted for this shared variance:

Given the description in the methods section, am I right in assuming that after preprocessing and the single-trial GLM, the authors then use the betas of the recall regressor in their further analyses, and that these betas represent only the variance uniquely explained by the recall (and not the recognition) regressor for all following analyses? Or in other words, the variance shared with the recognition regressor will in this GLM end up in the error term? I had trouble finding information about whether it is these recall betas that are used in later (e.g. decoding) steps of analysis, where the methods section mostly just refers to “neural activity” or “neural activation patterns”.

Previous point (6), asking whether data could be shown specifically for false alarm trials (related to Reviewer#2's point 7, who also asked for separate analyses for old and new recognition trials):

The authors' answer is not satisfying here. First, no new analyses were conducted to separately analyse old and new trials, as suggested by myself and Reviewer #2. Second, I do not agree with the statement that “The tendency for a given participant to rely on higher-level features can be measured by looking at the participants' average "lure" trial accuracy”. While reliance on high-level features would probably lead to higher lure false alarm rates, it is not possible, logically, to make an inference in the opposite direction (i.e., lure accuracy being any sort of specific index for dependence on high-level features). There are many possible reasons why participants produce high false alarm rates that have nothing to do with relying on higher-level features, including but not limited to a liberal response bias. The analysis presented in response to these comments, showing a brain-behaviour correlation only for those subjects with higher lure accuracy, does not sufficiently address the points raised.

Previous point (7), regarding the request to show a validation of the FSIC method on (cross-validated) perception data:

The answer here is not very convincing, stating that “FSIC might not function as intended when analysing perception data”, and that “Attention during perception/encoding may provide similar sources of variance [...] but this is unclear.” I still feel that the paper would benefit from showing a validation of the FSIC method for some other data than the recall periods, to provide the reader with a benchmark to compare the memory recall data against. This could be cross-validated perception data, as suggested in my previous comment. Or, if it is really impossible to perform the analysis on perception data, the recognition data (i.e. betas) from the memory trials could be used to show whether effects reported in the manuscript are specific to the recall data. The latter could potentially also address my above comment regarding correlated recall/recognition activity.

Reviewers' comments:

Reviewer #1 (Remarks to the Author):

This is the first revision of the manuscript „Feature-Specific Neural Reactivation during Episodic Memory“. The authors present various new analyses and interpretations of their results. I think the manuscript has been substantially improved, though some crucial points remain to be addressed.

Methods

1: The authors still need to show that their approach captures retinotopic representations. Their new results only show that an analysis with a higher resolution performs worse. They would need to show that activity at individual voxels provides significant information about the visual content at specific locations on the screen, and that adjacent voxels provide information about adjacent screen locations.

Our study is concerned with the representation of visual features, including feature location, and not retinotopic representations, per se. We were not sufficiently clear about this in previous version of the manuscript, and we appreciate the reviewer helping us clarify this point. To be more accurate with our language, we replaced “retinotopic” with “spatial” (page 11, line 221).

Our finding of significantly greater image classification accuracy using the original 3x3 features (i.e. containing some spatial information) vs. 1x1 features (i.e. containing no spatial information) indicates that the calcarine sulcus contains some spatial information during perception, and that our measure was able to capture that information.

Feature location is defined in relation to the image, not the point of fixation, so it is possible that the location information we decoded in the calcarine sulcus is represented in image-centric rather than fixation-centric coordinates, but this distinction is not something we aimed to address in the current study.

We should note, finally, that it is reasonable to expect a rough correspondence between image-centric and fixation-centric measures of location, because the participants' were focused on the image during encoding and recognition.

2: Here it would be great if the authors could provide some results – maybe from a new behavioral study – that validates their approach: after the rating, they could ask participants to describe as many details as possible. This would allow them to interpret these ratings more objectively.

We found that we could address this question without running a behavioral study. If vividness ratings are a valid index of memory quality, then higher vividness ratings should be associated with better accuracy on the recognition judgment. Note that the vividness ratings were made after the visualization period, but before the memory probe, so that participants did not know anything about the recognition probe at the time they made the vividness rating.

We have added a new results section “Relation Between Vividness Ratings and Old/New Task Accuracy” (page 9, line 171) and Supplementary Figure 1:

“In order to validate the subjective vividness ratings, we set out to determine if trials with higher ratings were associated with higher recognition accuracy. A significant positive relation between vividness and accuracy was found when all trials (“old” and “new”/lure) were included in the analysis [$\beta = .036$, $p < .001$]. Dividing the trials into “old” and “new”/lure conditions, a significant positive relation was found for “old” trials [$\beta = .058$, $p < .001$], whereas “new”/lure trials showed a marginal positive relation [$\beta = .021$, $p = .088$]. See Supplementary Figure 1 for more detail.”

10, 11: I am still not convinced that the network indeed produces meaningful results when presented with objects that are very different from the ones it has been trained on. The authors need to demonstrate that (and to which degree) there is indeed a semantic relationship between the objects they presented (including people etc.) and the labels the network assigns, which are a subset of those it has been trained on.

Here we have decided to run a small behavioral experiment to address the reviewer’s question. If the network does not produce meaningful results when the objects come from outside the training pool, then the predicted semantic categories should be mostly unrelated to the true semantics of the images used in our experiment.

Twelve participants (not including any participants from the main study) rated the accuracy of the top 5 (out of 1000) semantic labels predicted by VGG16 for the Encoding/Retrieval images. For a null reference, the labels were randomly shuffled (over the 180 images) for six subjects. There were four possible ratings: 1) correct classification, i.e. at least one of the labels was in the image (we were interested in general rather than specific semantic categories, e.g. if the image contained a dog then any dog breed label would be considered to be in the image); 2) at least one of the labels had a clear/direct semantic relation to the image; 3) at least one of the labels had a loose/indirect semantic relation to the image; 4) none of the labels had any semantic relation to the image. The results can be seen in Supplementary Figure 11.

“According to twelve independent raters, there are strong semantic relationships between training/test set images and the categorical labels VGG16 assigns: an average of 60% of the training/test set images had at least one label in the top 5 (out of 1000) that had a clear/direct semantic relation to the image—which was significantly greater than the 6% attained with shuffled labels ($t(10) = 12.7$, $p < .001$); see Supplementary Figure 11 for more details.” Added to page 37, line 801.

Results

3: The authors should test whether these accuracies are higher than would be expected by chance (i.e., higher than in label-shuffled surrogates)

We have added indications of significant time points to Supplementary Figure 2:

“Lines above the graph indicate time points with classification accuracy greater than chance; $p < .05$, one-tailed, FDR corrected. CIs and p values calculated by bootstrapping (1000 samples) over subjects’ mean reactivation values.”

Reviewer #2 (Remarks to the Author):

The authors were responsive to the comments that I and the other Reviewers raised. Some of the new analyses help address concerns that were raised, though other concerns still stand. For example, the data relating reactivation measures to behavior, while interesting, remain a bit underwhelming.

The authors have also better explained the FSIC method, but I still find the approach to be non-intuitive—or, better put, to be a non-direct measure of reactivation. The authors argue that using this informational connectivity approach helps address, for example, situations where subjects might remember the wrong image. They argue that, if subjects remember the wrong image on a given trial, traditional decoding would fail to detect any reactivation, but FSIC only cares about whether two regions have common information, whether or not that information is correct. Again, this is an indirect way to measure reactivation since the content of the reactivation is not actually verified (i.e., that it matches perception).

The obvious concern is that correlations between regions just reflect voxels within pairs of ROIs that have correlated activity (or noise correlations). The authors emphasize that if the correlations between regions were just noise correlations, then they would not differ as a function of the feature levels. This does seem to be an important point. While it still does not render the FSIC measure a “direct” measure of reactivation, I do agree that noise correlations should not differ by feature level. However, one point I am hoping the authors can clarify is exactly how non-target feature levels were regressed out. Obviously, it is critical that this regressing out of non-target feature levels isn’t what produces the selective FSIC measures for matching seed/feature levels. As an example, when considering the high-level ROI seed and the mid-level features, was the “target feature” the high-level features (i.e., relevant to the seed ROI) or was it the mid-level features? Depending on how this was done, the selectivity to the relevant feature level could simply be an artifact of regressing-out the non-target feature (but hopefully this wasn’t the case).

Thank you for helping us refine and clarify the description of our FSIC methodology. To answer the question, all feature levels in the target ROI were treated equally (as four independent variables in a LME model) for all four seeds, i.e. there was no bias towards the seed’s feature level in the LME model.

As described in the “Feature-Specific Informational Connectivity” methods section:

“For the FSIC analyses, partial regression coefficients were calculated (using trial-by-trial reactivation data from the recall period) with separate linear mixed-effects (LME) models for all combinations of seed ROI and target ROI. For each LME model, reactivation (rank measure) of the associated feature

level for the seed ROI was the dependent variable (DV), reactivation for each of the four feature levels within the target ROI were the independent variables (IV)..."

For each target ROI, the t-values depicted along each row of Figure 4b are the t-values of the coefficients for the four independent variables.

This lack of bias can be seen in the modeling results depicted in Supplementary Figure 4b. Here we see each seed correlating with all feature levels approximately equally (focused in the target ROIs that contain representations of the target feature level). This cross-feature-level correlation was caused by the trial-by-trial variation in the proportion of "remembered" features being identical across feature levels (in Supplementary Figure 4c the proportions are independent across feature levels).

Reviewer #3 (Remarks to the Author):

In the revision, Bone et al. made a number of changes to the manuscript, addressing many of the concerns raised by the reviews. These changes were mainly done by re-writing or addition sections, while few new datapoints or analyses were added since the first version.

The editor asked to comment on the issue of novelty raised by the previous reviews. I do agree that there is relatively little conceptual advance from this paper, in the sense that I did not learn anything substantially new about episodic memory or how memories are reactivated in the brain. For me, the main novelty lies in the proof of principle that CNNs are able to predict neural activity patterns during memory recall periods with little bottom-up input, and that the resulting mapping onto higher- and lower-level layers of the networks shows a meaningful relationship to behaviour. I believe that many memory researchers will find this useful and feel encouraged to use such models to test more specific predictions. I thus agree with the authors' conclusion that the results "show the potential for FSIC, and other feature-specific approaches that can decompose neural pattern representations, to test and elucidate the mechanisms [...]".

Many of my previous comments were sufficiently addressed in the revision. Below are those comments where I was not convinced by the authors' rebuttal.

Previous point (1), regarding the high correlation between the recall and recognition periods of a trial, and how the authors accounted for this shared variance:

Given the description in the methods section, am I right in assuming that after preprocessing and the single-trial GLM, the authors then use the betas of the recall regressor in their further analyses, and that these betas represent only the variance uniquely explained by the recall (and not the recognition) regressor for all following analyses? Or in other words, the variance shared with the recognition regressor will in this GLM end up in the error term? I had trouble finding information about whether it is these recall betas that are used in later (e.g. decoding) steps of analysis, where the methods section mostly just refers to "neural activity" or "neural activation patterns".

The reviewer is correct. We used the recall—and not recognition—regressor for all analyses addressing the feature reactivation. We have updated methods section to make this more clear.

“The “recall” beta coefficients were used in all subsequent neural analyses of the “imagery”/recall period (i.e. all neural analyses except for FSIC during the recognition period) and the “old/new judgment” beta coefficients were used in all subsequent neural analyses of the “old/new judgment”/recognition period (i.e. FSIC during the recognition period; see Supplementary Figure 6).” Added to page 35, line 744.

Previous point (6), asking whether data could be shown specifically for false alarm trials (related to Reviewer#2’s point 7, who also asked for separate analyses for old and new recognition trials):

The authors’ answer is not satisfying here. First, no new analyses were conducted to separately analyse old and new trials, as suggested by myself and Reviewer #2. Second, I do not agree with the statement that “The tendency for a given participant to rely on higher-level features can be measured by looking at the participants’ average “lure” trial accuracy”. While reliance on high-level features would probably lead to higher lure false alarm rates, it is not possible, logically, to make an inference in the opposite direction (i.e., lure accuracy being any sort of specific index for dependence on high-level features). There are many possible reasons why participants produce high false alarm rates that have nothing to do with relying on higher-level features, including but not limited to a liberal response bias. The analysis presented in response to these comments, showing a brain-behaviour correlation only for those subjects with higher lure accuracy, does not sufficiently address the points raised.

Thank you for this helpful remark. We have now done more extensive analyses on old and lure trials. The correlation between accuracy and feature-specific reactivation divided into old and lure trials is presented in Supplementary Figure 9. For both between- and within-subject analyses, low-level reactivation significantly correlated with old-trial accuracy rather than lure-trial accuracy (note: the within-subject correlation between accuracy and vividness was also largest for old trials; see results section “Relation Between Vividness Ratings and Old/New Task Accuracy” (page 9, line 171) and Supplementary Figure 1). Our results are consistent with the idea that the high-lure-accuracy subjects utilize recalled low-level details (e.g. edges) for the recognition task, but only when no difference in high-level/semantic features (between the subject’s memory and the probe) is clearly evident (which would happen more often on old trials, because there are no differences—assuming an accurate memory).

Note: Preliminary findings within the hippocampus for an upcoming study further validate our current approach: trial-by-trial accuracy positively correlated with reactivation of low-level visual details and negatively correlated with reactivation of semantic features in high-lure-accuracy subjects, whereas for low-lure-accuracy subjects the correlation with low-level visual details was negative and the correlation with semantic features was positive.

Previous point (7), regarding the request to show a validation of the FSIC method on (cross-validated)

perception data:

The answer here is not very convincing, stating that “FSIC might not function as intended when analysing perception data”, and that “Attention during perception/encoding may provide similar sources of variance [...] but this is unclear.” I still feel that the paper would benefit from showing a validation of the FSIC method for some other data than the recall periods, to provide the reader with a benchmark to compare the memory recall data against. This could be cross-validated perception data, as suggested in my previous comment. Or, if it is really impossible to perform the analysis on perception data, the recognition data (i.e. betas) from the memory trials could be used to show whether effects reported in the manuscript are specific to the recall data. The latter could potentially also address my above comment regarding correlated recall/recognition activity.

We have added a FSIC analysis of the recognition task data (betas estimated during recognition probe) as Supplementary Figure 6.

The most notable difference between the recognition and recall periods is the lack of low-level FISC within the lateral frontal cortex during recognition. This may be the result of the low-level features being generated in a predominately bottom-up manner during recognition (due to the perception of the probe), and a predominately top-down manner during recall (requiring lateral frontal regions for the construction and/or maintenance of the mental image in the visual cortex).

Reviewer #1 (Remarks to the Author):

All my remaining points have been convincingly addressed.

Reviewer #2 (Remarks to the Author):

In my prior review, I asked the authors to clarify how/whether feature levels were regressed out from various analyses. They have clarified this point and addressed the concern I raised. My concern about FSIC being a non-direct measure of reactivation still remains, but this is just inherent to the method and is not something I asked the authors to further address. Finally, as I had raised in my initial review--and as Reviewer 3 has raised in his/her reviews--the link between the reactivation measures and behavior are not very compelling. There are some reasonable, but relatively weak relationships. That said, I think the main take-away from this paper will be that CNN's can be used to index feature-specific information during memory retrieval. As such, the data linking reactivation to behavior are of secondary importance.

Reviewer #3 (Remarks to the Author):

The authors addressed the remaining major concerns in their second revision. I have two relatively minor comments, see below. In addition, I strongly encourage the authors to share at minimum their code (and ideally data too) upon publication of this manuscript. Otherwise, given the complexity of the methods, it will be impossible for other scientists to reproduce the results.

(1) Regarding the FSIC analysis of the recognition probe phase (added in the most recent revision), the authors should compute the correlations shown in Fig. 5 also for the recognition data, so that the reader can determine whether these brain-behaviour correlations are specific to the recall part of a trial, or more generic in nature. This is important since many of the conclusions of the paper hinge on these correlations, and their specificity to episodic recall.

(2) The statement in the abstract that the results of the manuscript “clarify the role of frontal cortex during episodic recall” seems overstated. The results do show that lower-level features are represented in frontal cortex specifically during recall (and not recognition), which is an interesting observation, but not a clarification of the role of the frontal cortex during recall. In fact, no conclusion regarding frontal cortex is offered in the “conclusions” section of the discussion, and the authors should thus attenuate this statement in the abstract.

****REVIEWERS' COMMENTS:**

Reviewer #1 (Remarks to the Author):

All my remaining points have been convincingly addressed.

No request.

Reviewer #2 (Remarks to the Author):

In my prior review, I asked the authors to clarify how/whether feature levels were regressed out from various analyses. They have clarified this point and addressed the concern I raised. My concern about FSIC being a non-direct measure of reactivation still remains, but this is just inherent to the method and is not something I asked the authors to further address. Finally, as I had raised in my initial review--and as Reviewer 3 has raised in his/her reviews--the link between the reactivation measures and behavior are not very compelling. There are some reasonable, but relatively weak relationships. That said, I think the main take-away from this paper will be that CNN's can be used to index feature-specific information during memory retrieval. As such, the data linking reactivation to behavior are of secondary importance.

FSIC (a measure of feature-specific information shared between brain regions) is intended as the focus of the paper.

With respect to the link between feature-specific reactivation and behavior, the relation between feature-specific neural reactivation and vividness is clear, as is the between-subject correlation between feature-specific neural reactivation and accuracy. The within-subject correlation between feature-specific neural reactivation and accuracy was significant for subjects with high lure accuracy and consistent with our expectations and the between-subject result. As stated in the paper, the fact that the within-subject result was limited to the high lure accuracy participants is itself an interesting finding that suggests a meaningful individual difference in the representation/utilization of low-level features during recognition. An upcoming paper from our lab on feature-specific reactivation in the hippocampus supports the above findings of an individual difference in the representation of low-level features and expands on them. Specifically, we have found evidence that the individual difference is due to the low-lure-accuracy subjects having lower memory accuracy (i.e. memories tend to be further away from perception in feature-space) which causes the representations of low-level features with small receptive fields to hinder, rather than facilitate, recognition.

Reviewer #3 (Remarks to the Author):

The authors addressed the remaining major concerns in their second revision. I have two relatively minor comments, see below. In addition, I strongly encourage the authors to share at minimum their code (and ideally data too) upon publication of this manuscript. Otherwise, given the complexity of the methods, it will be impossible for other scientists to reproduce the results.

(1) Regarding the FSIC analysis of the recognition probe phase (added in the most recent revision), the authors should compute the correlations shown in Fig. 5 also for the recognition data, so that the reader can determine whether these brain-behaviour correlations are specific to the recall part of a trial, or more generic in nature. This is important since many of the conclusions of the paper hinge on these correlations, and their specificity to episodic recall.

Added analyses as Supplementary Figure 10. Referenced in the article on page 16, line 326.

With regard to the point in the last sentence, we anticipated that much of the recalled information/reactivation would be carried over to the recognition phase to facilitate performance. This is one reason we expected correlations between reactivation during recall and recognition accuracy. As noted in the legend of Supplementary Figure 10, we didn't originally include the requested correlations because patterns of neural activation caused by perception of the recognition probe would be conflated with (and consequently obscure) memory-driven neural reactivation—particularly for low-level features within the early visual cortex. Our main findings used reactivation during the recall period to avoid this issue.

(2) The statement in the abstract that the results of the manuscript “clarify the role of frontal cortex during episodic recall” seems overstated. The results do show that lower-level features are represented in frontal cortex specifically during recall (and not recognition), which is an interesting observation, but not a clarification of the role of the frontal cortex during recall. In fact, no conclusion regarding frontal cortex is offered in the “conclusions” section of the discussion, and the authors should thus attenuate this statement in the abstract.

Statement removed.